# Diagnosing the radiative and chemical contributions to future changes in tropical column ozone with the UM-UKCA chemistry-climate model

James Keeble[1], Ewa M. Bednarz[1], Antara Banerjee[2], N. Luke Abraham[1,3], Neil R. P. Harris[4], Amanda C. Maycock[5] and John A. Pyle[1,3]

[1]University of Cambridge, Department of Chemistry, Cambridge, UK
[2]Department of Applied Physics and Applied Mathematics, Columbia University, New York, NY, USA
[3]NCAS/University of Cambridge, Department of Chemistry, Cambridge, UK
[4]Centre for Atmospheric Informatics and Emissions Technology, Cranfield University, Cranfield, UK
[5]School of Earth and Environment, University of Leeds, Leeds, UK

*Correspondence to*: J. Keeble (james.keeble@atm.ch.cam.ac.uk)

**Abstract.** Chemical and dynamical drivers of trends in tropical total column ozone (TCO3) for the recent past and future periods are explored using the UM-UKCA chemistry-climate model. A transient 1960-2100 simulation is analysed which follows the representative concentration pathway 6.0 (RCP6.0) emissions scenario for the future. Tropical averaged (10°S-10°N) TCO3 values decrease from the 1970s, reach a minimum around 2000, and return to their 1980 values around 2040, consistent with the use and emission of halogenated ozone depleting substances (ODS), and their later controls under the Montreal Protocol. However, when the ozone column is subdivided into three partial columns (PCO3) that cover the upper stratosphere (PCO3$_{US}$), lower stratosphere (PCO3$_{LS}$) and troposphere (PCO3$_T$), significant differences in the temporal behaviour of the partial columns is seen. Modelled PCO3$_T$ values under the RCP6.0 emissions scenario increase from 1960-2000 before remaining approximately constant throughout the 21$^{st}$ century. PCO3$_{LS}$ values decrease rapidly from 1960-2000, remain constant from 2000-2050, before gradually decreasing further from 2050-2100, never returning to their 1980s values. In contrast, PCO3$_{US}$ values decrease from 1960-2000, before increasing rapidly throughout the 21$^{st}$ century, returning to 1980s values by ~2020, and reach significantly higher values by 2100. Using a series of idealised UM-UKCA time-slice simulations with concentrations of well-mixed greenhouse gases (GHG) and halogenated ODS species set to either year 2000 or 2100 levels, we examine the main processes that drive the PCO3 responses in the three regions, and assess how these processes change under different emission scenarios. Finally, we present a simple, linearised model to describe the future evolution of tropical stratospheric column ozone values based on terms representing time-dependent abundances of GHG and halogenated ODS.

# 1 Introduction

Total column ozone (TCO3) has a direct effect on human health by preventing harmful ultraviolet (UV) radiation from reaching the surface. It is therefore important to gain a quantitative understanding of how TCO3 values may evolve over the 21st century. While ozone mixing ratios are on average highest in the tropical stratosphere, tropical TCO3 values are the
lowest of any region outside of the Antarctic ozone hole (World Meteorological Organization (WMO), 2014), due in part to the maximum ozone mixing ratios being found at higher altitudes in the tropics and the tropopause height being higher there than in mid and high latitudes. This fact, combined with the high population of many tropical countries, means it is important to understand the various factors that will affect TCO3 values over the course of the 21st century.

The discovery of the ozone hole by Farman et al. (1985) ultimately led to controls on the emissions of CFCs and other
ozone-depleting substances (ODS) through the Montreal Protocol and its subsequent Adjustments and Amendments (WMO, 2014). As a result, stratospheric concentrations of inorganic chlorine are expected to decline throughout the 21st century (e.g. Mäder et al., 2010), and stratospheric ozone concentrations in the mid- and high latitudes are projected to return to their pre-1980s values (Eyring et al., 2013a; WMO, 2014). However, future projections of tropical TCO3 abundances show a large inter-model range (e.g. Austin et al., 2010; WMO, 2011; 2014), with recent studies indicating that tropical TCO3 may
not return to pre-1980s values by the end of the 21st century despite reduction in stratospheric halogenated ODS concentrations (e.g. Eyring et al., 2013a; Meul et al., 2016).

In the extra-polar stratosphere, local ozone concentrations are determined by the balance between production and destruction of ozone through gas phase chemical reactions, plus transport into and out of the region of interest (e.g. Brewer and Wilson, 1968; Garny et al., 2011). $O_x$ mixing ratios (where $O_x$, or odd oxygen, is defined as the sum of ozone ($O_3$) and atomic
oxygen (O)) are determined by sets of photochemical reactions first described by Chapman (1930) plus ozone destroying catalytic cycles involving chlorine, nitrogen, hydrogen and bromine radical species (e.g. Bates and Nicolet, 1950; Crutzen, 1970; Johnston, 1971; Molina and Rowland, 1974; Stolarski and Cicerone, 1974). Unlike in the polar lower stratosphere, heterogeneous processes play only a minor role in determining tropical TCO3 abundances, although this can change after large volcanic eruptions (e.g. Solomon et al., 1996; Telford et al., 2009), in association with aerosol transport within the
Asian summer monsoon circulation (Solomon et al., 2016), or as a result of proposed stratospheric aerosol geoengineering schemes (e.g. Weisenstein et al., 2015; Tang et al., 2016).

Changes in anthropogenic emissions during the 21st century are expected to perturb stratospheric ozone chemical cycles involving $O_x$, $ClO_x$ (Cl+ClO), $NO_x$ (NO+NO$_2$) and $HO_x$ (OH+HO$_2$) in two ways. Firstly, the radiative effects of well-mixed GHGs affect both gas phase kinetics and stratospheric dynamics. Secondly, some GHGs also act as source gases for reactive
species: CFCs are source gases for inorganic chlorine ($Cl_y$), $N_2O$ is a source gas for reactive nitrogen ($NO_y$) and $CH_4$ is a source gas for $HO_x$.

Cooling of the stratosphere due to increased GHG concentrations, particularly $CO_2$, increases stratospheric ozone concentrations through both increases in the rate constant for the reaction $O+O_2+M$, leading to an increase of the ratio of $O_3$ to $O$, and decreases in the rate constant for the reaction $O+O_3$ (e.g., Barnett et al., 1974; Haigh and Pyle, 1982; Jonsson et al., 2004). In a similar way, the rate constants for the catalytic loss cycles involving $NO_x$, $HO_x$ and $ClO_x$ radicals are also temperature dependent (e.g. Brasseur and Hitchman 1988; Randeniya et al., 2002; Rosenfield et al., 2002; Stolarski et al., 2015), and so the combined efficiency of these cycles for destroying stratospheric ozone is also affected by GHG-induced stratospheric temperature changes.

Changes to emissions of CFCs, $N_2O$ and $CH_4$ will alter the concentrations of $ClO_x$, $NO_x$, and $HO_x$ radicals, affecting the catalytic cycles that destroy ozone (e.g., Chipperfield and Feng, 2003; Ravishankara et al., 2009). While future stratospheric halogen loadings are expected to decrease throughout the 21st century, emissions of $CH_4$ and $N_2O$, which are not regulated in the same way as halogenated ODS, are associated with greater uncertainty. The atmospheric concentration of these species, and by extension future concentrations of $HO_x$ and $NO_x$ radicals, is therefore highly sensitive to assumptions made about their future emissions.

The physical climate response to increases in GHG concentrations is expected to include increasing tropopause height, an acceleration of the Brewer-Dobson Circulation (BDC), and changes in the width of the region of the tropical upwelling in the lower stratosphere (e.g. Butchart et al., 2006, 2010; Garcia et al., 2007; Lorenz and DeWeaver, 2007; Shepherd, 2008; Li and Austin, 2008; Shepherd and McLandress, 2011; Hardiman et al., 2014; Palmeiro et al., 2014). Changes in the strength of the BDC affect ozone concentrations by directly transporting ozone out of the lower stratosphere (e.g. Plumb, 1996; Avallone and Prather, 1996), and by controlling the abundance of $Cl_y$, $NO_y$ and $HO_x$, which determines the chemical processing of ozone (e.g. Revell et al., 2012; Meul et al., 2014). In addition to the mean advection of air masses, quasi-horizontal mixing along isentropes is also important for the transport of stratospheric chemical constituents (Hall and Waugh, 1997). However, in the tropics horizontal mixing is relatively weak due to the existence of a sub-tropical transport barrier, the tropical pipe, which acts to some extent to isolate the tropical lower stratosphere from the mid latitudes (Waugh 1996; Neu and Plumb, 1999).

Since the photochemical lifetime of ozone is long in the lower stratosphere and short in the upper stratosphere, the relative importance of the chemical and dynamical processes described above will vary with altitude, with dynamical changes playing a more important role for ozone in the lower stratosphere and gas phase chemistry a more important role in the upper stratosphere. This makes it challenging to understand the sources of uncertainty and inter-model differences in future tropical TCO3 trends (e.g. WMO, 2014).

Alongside changes to stratospheric ozone concentrations, tropospheric ozone abundances are projected to change throughout the 21st century due to changes in future emissions of anthropogenic and natural species, particularly ozone precursors (e.g.

CO, $CH_4$, $NO_x$ and VOCs) and changes in climate (e.g. Eyring et al., 2013a; Banerjee et al., 2016; Meul et al., 2016). Changes to emissions of ozone precursors directly affect tropospheric ozone concentrations by affecting chemical production through reactions between $NO_x$, hydrocarbons and CO, which account for ~90% of local ozone production (Denman et al., 2007). While changes to ozone precursors are not considered in this study, changes in climate can affect tropospheric ozone abundances by changing water vapour, lightning $NO_x$ emissions ($LNO_x$) and stratosphere-troposphere exchange of ozone (STE) (e.g. Thompson et al., 1989; Eyring et al., 2013a; Young et al., 2013; Revel et al., 2015; Banerjee et al., 2016). These changes are an important consideration when assessing tropical TCO3 trends resulting from changes in GHG and halogenated ODS. It is important to note that while stratospheric column ozone prevents harmful UV radiation reaching the surface, tropospheric ozone is associated with a number of harmful effects on human health, air quality and the environment as it is an air pollutant and GHG (e.g. West et al., 2007; Revell et al., 2015). Therefore, any benefits related to increases in TCO3 resulting from increased tropospheric partial column values could be offset by the negative effects of increased surface ozone concentrations.

To assess the impacts of future anthropogenic emissions on atmospheric chemistry and climate, a number of representative concentration pathway (RCP) scenarios based on different assumptions about future socio-economic development have been developed (van Vuuren et al., 2011). While stratospheric chlorine loadings are predicted to decrease in the future in all RCP emissions scenarios, emissions of $CO_2$, $CH_4$ and $N_2O$ are associated with greater uncertainty and hence follow a wider range of pathways between the different RCP scenarios (WMO, 2011, 2014; IPCC, 2013; Meinshausen et al., 2011). For example, $CH_4$ and $N_2O$ emissions are projected to decline during the 21st century in RCP2.6, peak around the year 2040/2080 in RCP4.5/RCP6.0, respectively, and increase monotonically throughout the century in RCP8.5. The multitude of drivers and processes that affect atmospheric ozone abundances motivates the use of chemistry-climate models (CCMs) to explore changes in TCO3 over the 21st century under the different RCP scenarios (e.g. Eyring et al., 2013a; Iglesias-Suarez et al., 2016).

Here we present results of a modelling study that assesses projected trends in tropical column ozone. The aims of this paper are to: 1) analyse separately the contributions from different altitude regions to future tropical column ozone trends; 2) quantitatively determine the major chemical and physical drivers of the modelled partial tropical column ozone trends; and 3) formulate a simple model to estimate future tropical stratospheric column ozone changes and the contribution from the key drivers identified in 2) to these changes. The emphasis here is on the impact of halogenated ODS and the climatic effects of well-mixed GHGs on ozone chemistry and transport. We therefore do not consider the chemical effects of future $N_2O$ and $CH_4$ emissions, which will also contribute to future tropical column ozone trends (e.g. Butler et al., 2016; Revell et al., 2012) and show differences in their future concentrations across RCP scenarios (Meinshausen et al., 2011). We recognize that the future evolution of tropical ozone will depend, *inter alia*, on the ODS, GHG and tropospheric ozone precursor emissions scenario. Some of these are regulated, some are not and some will respond to climate change. Accordingly, the aim is not to

predict the precise evolution of tropical column ozone, but rather explore the contributions from the drivers stated above to future changes over a particular sub-set of scenarios. By breaking down our analysis into different vertical regions within which ozone levels are governed by fundamentally distinct processes, we aim to develop some general understanding of the processes that will affect tropical column ozone throughout the 21st century.

Section 2 describes the CCM simulations used for this study. In section 3, the modelled column ozone trends are discussed and separated into contributions from the upper stratosphere, lower stratosphere and troposphere, before the key drivers of column ozone trends in these separate altitude regions are discussed in section 4. In section 5, we produce a simple linear model to describe future tropical stratospheric column ozone changes as a function of GHG and halogenated ODS concentrations. Finally, the results are summarized in section 6.

**2 Model setup and experimental design**

For this study, we use version 7.3 of the Met Office's Unified Model HadGEM3-A (Hewitt et al., 2011) coupled with the United Kingdom Chemistry and Aerosol scheme (hereafter referred to as UM-UKCA). The model is run in atmosphere-only mode with a horizontal resolution of 2.5° latitude by 3.75° longitude, 60 vertical levels up to 84 km, and prescribed sea surface temperatures and sea ice extents. For this study, two configurations of UM-UKCA were used which are described
below.

We use an ensemble of transient simulations following the experimental design of the IGAC/SPARC CCMI REF-C2 experiment, which adopts the RCP6.0 scenario for future GHG and WMO (2011) recommendations for ODS concentrations (Eyring et al., 2013b). These simulations were performed using a configuration of UKCA with an extended stratospheric chemistry scheme to that described by Morgenstern et al. (2009), in which halogen source gases are considered explicitly, resulting in an additional 9 species, 17 bimolecular and 9 photolytic reactions. The tropospheric chemistry scheme in this
configuration of UKCA is relatively simplified and includes the oxidation of a limited range of organic species (CH$_4$, CO, CH$_3$O$_2$, CH$_3$OOH, HCHO) alongside detailed HO$_x$ and NO$_x$ chemistry. This configuration of UKCA was used for the recent SPARC Report on the Lifetimes of Stratospheric Ozone-Depleted Substances, their Replacements and Related Species (SPARC 2013; Chipperfield et al., 2014) and is described in detail in Bednarz et al. (2016). The model is forced at the lower
boundary with sea surface temperatures and sea ice fields taken from a coupled atmosphere-ocean HadGEM2-ES integration (Jones et al., 2011). In total, four ensemble members are used in this study: two integrations run from 1960-2099, and two integrations run from November 1980 to December 2080. The latter two ensemble members were initialised using different atmospheric initial conditions taken from a supporting perpetual year 1980 integration. The four ensemble members have identical time-dependent boundary conditions, thereby providing an estimate of the contribution from internal atmospheric
variability to simulated temporal variability and trends. All transient integrations used in this study include the effects of the 11-year solar cycle in both the radiation and photolysis schemes.

The transient simulations described above include both the radiative and chemical effects of time-varying anthropogenic source gases, specifically $CO_2$, $CH_4$, $N_2O$ and halogenated ODS. In order to separate the relative radiative and chemical contributions to future tropical ozone differences, the transient simulations were supplemented by time-slice integrations performed using a configuration of UKCA with a coupled stratosphere-troposphere chemistry scheme as described by

Banerjee et al. (2014). This scheme includes a more detailed tropospheric chemistry scheme (O'Connor et al., 2014) and the original UM-UKCA stratospheric chemical scheme described by Morgenstern et al. (2009). Six time-slice experiments were performed with this configuration of UM-UKCA that include different prescribed SSTs and sea ice, GHG and halogenated ODS concentrations (Banerjee et al., 2016). These include a set of simulations in which the physical climate state alone (e.g. SSTs, sea ice, radiative effects of GHG concentrations) is perturbed from a year 2000 baseline to year 2100 conditions taken

from either the RCP4.5 or RCP8.5 scenario. Note that when perturbing the physical climate state, GHG concentrations are not perturbed in the chemistry scheme i.e. the chemical impacts of changing $N_2O$, $CH_4$ and CFCs are not considered. The chemical effects of ODS in particular are considered as a separate perturbation: in the chemistry scheme, each pair of experiments in turn uses halogenated ODS loadings for either the year 2000 or 2100. The RCP4.5 scenario is used to determine the year 2100 halogenated ODS levels, although the exact scenario followed is arbitrary since all RCPs show

similar projections for future ODS emissions (Meinhausen et al., 2011). The resulting set of time-slice experiments are named accordingly to reflect the climate condition and chemical ODS loadings, e.g. TS2000$_{ODS}$ includes year 2000 climate conditions and year 2100 chemical ODS loadings, while TS4.5 includes year 2100 climate conditions following the RCP4.5 scenario and year 2000 ODS loadings (see Table 1). In all time-slice experiments chemical concentrations of $N_2O$ and $CH_4$ use prescribed year 2000 concentrations from RCP6.0 as a lower boundary condition, and thus their chemical effects are not

considered in this study. In principle, further time-slice experiments could be performed to also explore the chemical impacts of changes in tropospheric ozone precursors, unregulated short-lived halogen compounds and $N_2O$ and $CH_4$ changes. However, owing to limitations in computational resource we focus our attention on the effects of ODS and GHG-driven changes in climate. Each UM-UKCA time-slice experiment was run for 20 years, with the first 10 years discarded as spin-up for the model.

The design of the time-slice experiments allows for a quantitative separation of the radiative and chemical effects of some of the known drivers of stratospheric ozone changes over the 21[st] century, which can then aid in the interpretation of the simulated time-dependent changes in tropical column ozone in the transient integrations. We purposefully use time-slice experiments with different combinations of forcings to those in the transient simulations (time-slices are run for RCP 4.5 and 8.5 while the transient simulation is run for RCP 6.0), so that we can assess linearities in the ozone response to both ODS

and GHG changes.

Throughout the remainder of this study the impact of changing GHG concentrations is expressed in terms of differences in Carbon Dioxide Equivalent (CDE; IPCC, 2007), while ODS will be used to refer only to the halogenated ozone depleting

substances, and does not include $N_2O$, itself an important ozone depleting substance (e.g. WMO, 2014; Ravishankara et al., 2009). ODS concentrations have been calculated using the equivalent stratospheric chlorine (ESC) definition of Eyring et al. (2007), where $ESC = Cl_y + \alpha Br_y$, and $\alpha = 60$.

## 3 Modelled column ozone trends

The analysis presented in this study focuses on area weighted averages over 10°S-10°N. While previous studies of tropical ozone trends have used a broader region to define the tropics, typically from 25°S-25°N (e.g. Austin et al., 2010; Eyring et al., 2010; Meul et al., 2014), Hardiman et al. (2013) show that, in an ensemble of CMIP5 models following the RCP8.5 scenario, as the magnitude of the tropical upwelling mass flux is projected to increase over the 21[st] century, the width of the region of upwelling narrows at altitudes below 20 hPa. In order to avoid the impacts of changes to the width of the region

of tropical upwelling resulting from increases in GHG concentrations, in this study we use a narrower definition of the tropics. However, the results presented in this study were not changed significantly when a broader definition of the tropics (30°S-30°N) was used.

In this section, we first describe the changes in tropical total column ozone (defined herein as 0-48 km) and then the partial column trends for the upper and lower stratosphere and the troposphere. The processes driving these changes are then

explored in Section 4.

### 3.1 Total column ozone differences

Figure 1 shows tropical averaged TCO3 anomalies relative to a baseline period of 1995-2005 from 1960 to 2100 for each individual ensemble member (grey lines) and the ensemble mean 11-year running mean (black line). The baseline period 1995-2005 is chosen so that the transient simulations can be directly compared to the year 2000 time-slice experiments (see

Section 2). Also shown in Figure 1 are tropical averaged TCO3 anomalies from version 2.8 of the Bodeker Scientific total column ozone dataset (purple line; Bodeker et al., 2005). There is generally good agreement between the modelled tropical column ozone anomaly values and the Bodeker dataset, specifically with regards to the long-term changes during the period the model and observations overlap, and the magnitude of interannual variability. The ensemble mean 11-year running mean TCO3 abundances in the transient UM-UKCA simulation are generally anti-correlated with the long-term changes in

stratospheric chlorine levels, consistent with other studies (e.g. Eyring et al., 2013a). There is a decline in tropical TCO3 of ~6 DU from the mid-1970s to 1990, coincident with increases in stratospheric $Cl_y$ concentrations resulting from the emission of halogenated ODSs. TCO3 values remain relatively low from 1990 to 2010, before gradually returning to 1980s values by ~2040 and to 1960s values by ~2050, after which they remain relatively constant from 2050-2090. Beyond 2090 there is evidence for a further decrease, bringing column values once again below their 1980s values. This behaviour is broadly

consistent with previous studies (e.g. Oman et al., 2010; Eyring et al., 2013a; Meul et al., 2014), with the main exception

being that while other studies show an increase in tropical TCO3 over the first half of the 21$^{st}$ century, they do not generally indicate a return to 1980s values. Results from the UM-UKCA transient simulations show that tropical TCO3 values may return to pre-1980s values for part of the 21$^{st}$ century, but by the end of the century will begin to decrease again. Superimposed on the TCO3 11-year running mean is the signal of the 11-year solar cycle, which leads to variations of <5 DU between solar maximum and minimum. The large degree of natural variability simulated in the model highlights the difficulties in assessing ozone trends and return dates from relatively short observational records (as discussed by Harris et al., 2015).

The time-slice experiments, plotted in Figure 1 as discrete points (circles and triangles), show the dependence on the RCP scenario of modelled year 2100 tropical TCO3 values. Tropical TCO3 increases by around 12 DU when stratospheric $Cl_y$ loadings are decreased from their year 2000 values to year 2100 values under fixed year 2000 GHG conditions (TS2000 - TS2000$_{ODS}$, shown by the difference between the green symbols in Figure 1). However, the same decrease in stratospheric $Cl_y$ abundances leads to slightly smaller increases in tropical TCO3 when future changes in climate are also included according to the RCP4.5 or RCP8.5 scenario (11 DU for TS4.5 – TS4.5$_{ODS}$ and 10 DU for TS8.5 – TS8.5$_{ODS}$, shown by the differences between the blue and red symbols in Fig. 1, respectively). We explore the processes controlling these changes in Section 4.

The effect on tropical TCO3 of future climatic changes resulting from increases in GHGs is seen by comparing the TS4.5 and TS8.5 time-slice integrations with TS2000. Under a more moderate increase in GHG concentrations (TS4.5 - TS2000, compare green and blue circles Figure 1), tropical TCO3 increases by 4.0 DU between year 2000 and 2100, while under a much larger GHG concentration change (TS8.5 - TS2000, compare green and red circles in Figure 1), tropical TCO3 values show no change, indicating a non-linear response to the magnitude of GHG forcing (Banerjee et al., 2016). The causes of this are discussed further in Section 4.

### 3.2 Partial column ozone differences

Projected trends in tropical ozone concentrations show a complex vertical structure (WMO, 2014). In this section we assess modelled changes in tropical ozone partial columns for the upper stratosphere (30-48 km; PCO3$_{US}$), lower stratosphere (tropopause to 30 km; PCO3$_{US}$) and troposphere (PCO3$_T$). The 30 km boundary between the lower and upper stratosphere corresponds to an approximate pressure of 15 hPa. This level is chosen as an approximation for the transition region between ozone being predominantly under photochemical control in the upper stratosphere and predominantly under dynamical control in the lower stratosphere. Note that we take into account any changes in tropopause height, as defined by the lapse rate tropopause (WMO, 1957), when calculating the partial columns within each experiment, rather than using a fixed altitude. Thus, any changes in tropopause height will affect the lower stratosphere and tropospheric partial columns even if the vertical distribution of ozone concentrations is unchanged.

PCO3$_{US}$ values decrease by around 4 DU from 1960 to the late 1990s (Figure 2a), consistent with the increasing stratospheric Cl$_y$ concentrations over this period. From around 2000 onwards, PCO3$_{US}$ values increase rapidly due to a combination of decreased stratospheric Cl$_y$ concentrations and the GHG-induced stratospheric cooling effect, returning to 1980 values by ~2020, and 1960 values by ~2040. From 2040, PCO3$_{US}$ values continue to increase to around 3-4 DU above their 1960s values by 2100 – the well-known ozone "super recovery" effect (Chipperfield and Feng, 2003).

The time-slice experiments show that an increase in PCO3$_{US}$ values of ~5 DU can be attributed to Cl$_y$ changes over the 21$^{st}$ century (calculated as the difference between the green symbols in Figure 2a), although the exact magnitude of this increase is dependent on the background climate, as discussed above. As well as responding to changes in stratospheric Cl$_y$, PCO3$_{US}$ values in the late 21$^{st}$ century are dependent on the GHG emissions scenario, since $CO_2$ is the main driver of stratospheric cooling (e.g. Manabe and Wetherald, 1975; Shine et al., 2003). The TS4.5 and TS8.5 experiments, which consider only the differences in physical climate resulting from GHG increases, both show higher TCO3 values than TS2000 (+5 DU for TS4.5, comparing green and blue circles in Fig. 2a, and +12 DU for TS8.5, comparing green and red circles). These results can be used to calculate an approximate change in tropical PCO3$_{US}$ per unit change in CDE of $\frac{\Delta PCO3_{US}}{\Delta CDE} \approx 0.02$ DU ppmv$^{-1}$ and per unit change in ESC of $\frac{\Delta PCO3_{US}}{\Delta ESC} \approx -1.72$ DU ppbv$^{-1}$. These relationships indicate that over the recent past upper stratospheric ozone depletion resulting from increased Cl$_y$ concentrations has in part been offset by radiative cooling resulting from increased GHG concentrations (consistent with Shepherd and Jonsson, 2008), and that in the future both increased GHG concentrations and reduced stratospheric Cl$_y$ will result in increases in upper stratospheric ozone concentrations. However, as discussed for TCO3, the impact of ODS changes on upper stratospheric partial column abundance is dependent on GHG concentrations (compare blue/red circles with blue/red triangles in Figure 2a).

As was found in the upper stratosphere, the modelled historical trend in PCO3$_{LS}$ is strongly negative, with a decrease of ~6 DU from 1960 to the late 1990s (Figure 2b). However, the projected future trend in PCO3$_{LS}$ differs greatly from that in the upper stratosphere. From 2000 to ~2050, modelled PCO3$_{LS}$ abundances remain approximately steady, before decreasing during the latter half of the 21$^{st}$ century.

The time-slice experiments demonstrate the competing effects of decreasing stratospheric Cl$_y$ and changes in physical climate from increasing GHG concentrations on PCO3$_{LS}$ over the 21$^{st}$ century. As in the upper stratosphere, projected decreases in stratospheric Cl$_y$ result in an increase in PCO3$_{LS}$ between year 2000 and 2100 (compare green triangle and circle in Figure 2b). However, changes to the physical climate from increased GHG concentrations lead to decreases in PCO3$_{LS}$ (compare blue/red circles with green circle in Figure 2b). For changes in GHGs alone, the magnitude of the PCO3$_{LS}$ response increases with the magnitude of the CDE perturbation (-4 DU for TS4.5-TS2000, -16 DU for TS8.5-TS2000). As for upper stratospheric partial column values, changes to tropical PCO3$_{LS}$ values per unit change in CDE and ESC

concentrations can be calculated, giving $\frac{\Delta PCO3_{LS}}{\Delta CDE} \approx$ -0.03 DU ppmv$^{-1}$ and $\frac{\Delta PCO3_{LS}}{\Delta ESC} \approx$ -1.92 DU ppbv$^{-1}$. While increases in both GHGs and stratospheric $Cl_y$ have acted to decrease $PCO3_{LS}$ in the past, in the future the effects of decreasing stratospheric $Cl_y$ and increasing GHG concentrations will have competing effects on $PCO3_{LS}$. This is in contrast to the upper stratosphere where future decreases in halogenated ODS and increases in GHG concentrations are both projected to lead to higher ozone concentrations. As was seen for the upper stratosphere, the $PCO3_{LS}$ response to a given change in ODS is also dependent on the GHG concentration, (+7 DU for TS2000$_{ODS}$ - TS2000, +6 DU for TS4.5$_{ODS}$ - TS4.5 and +4 DU for TS8.5$_{ODS}$ - TS8.5, see Figure 2b).

From 1960 to 2000, the tropical tropospheric partial ozone column ($PCO3_T$) increases by approximately 5 DU (Figure 2c), then remains constant until 2040, before increasing again by ~2 DU by 2060. There is some suggestion from the transient simulations that for the RCP6.0 emissions scenario tropical $PCO3_T$ values decrease to year 2000 values during the final decade of the 21$^{st}$ century. The time-slice experiments indicate that tropical $PCO3_T$ values are relatively insensitive to changes in the physical climate state alone for the two RCP scenarios considered here, with values in TS4.5 and TS8.5 both increasing by ~5 DU. As expected, tropical $PCO3_T$ shows no significant response to changes in ODS concentrations irrespective of the GHG loading, as the long-lived halogenated ODS species are not oxidised until they reach the stratosphere. However, we remind the reader that by design the time-slice simulations do not explore the chemical roles of tropospheric $CH_4$, $NO_x$ and volatile organic compounds (VOCs) emissions, which are likely to be important drivers of tropospheric ozone changes in the transient simulations; this is discussed further in Section 4.3.

## 4 Drivers of column ozone changes

The results in Section 3 show that changes in stratospheric $Cl_y$ and the physical climate effects of GHGs have distinct impacts on partial column ozone in different altitude ranges. These behaviours reflect the various chemical and transport processes that form the dominant control on tropical ozone abundances in different regions. In the following sections the major mechanisms operating in each of the three partial column regimes are explored.

### 4.1 Upper Stratosphere

As discussed in Section 3.2, $PCO3_{US}$ values are projected to return quickly to pre-1980 values over the next few decades and continue to increase throughout the 21$^{st}$ century, leading to super-recovery of $PCO3_{US}$ by 2100 (Figure 2a). This can be seen further in Figure 3, which shows annual mean $PCO3_{US}$ values plotted as a function of stratospheric ESC at 45 km, for both the time-slice and transient simulations. Data for the transient integrations covering 1960-2100 are shown as crosses, with different colours denoting different 20 year periods. From 1960 to 2000, as ESC concentrations rapidly increase by ~3 ppbv, $PCO3_{US}$ abundances decrease by ~3 DU. From 2000-2100, as ESC concentrations decrease, $PCO3_{US}$ abundances are projected to increase, but the trend from 2000 to 2100 does not retrace the trend from 1960 to 2000. Instead, the transient

integrations indicate a larger change in PCO3$_{US}$ per unit change in ESC in the future compared to over the past owing to the higher background GHG concentrations.

The time-slice experiments can be used to quantify the separate and combined effects of GHG-induced changes in climate and the chemical effects of ODS on PCO3$_{US}$ changes. The upper rows in Table 2 give values for the TS2000, TS2000$_{ODS}$, TS4.5 and TS8.5 simulations of PCO3$_{US}$ abundances, chemical loss of O$_x$ through reactions with each of the key chemical families (halogens, HO$_x$, NO$_x$, and Ox), chemical production and O$_x$ lifetime. Chemical loss of O$_x$ is calculated following Lee et al. (2002) with the total rate of O$_x$ destruction calculated as the sum of the rates of each chemical ozone loss cycle included in the model chemical scheme. As discussed above, N$_2$O and CH$_4$ concentrations are kept constant in the chemical scheme in all time-slice simulations, and thus any change in NO$_x$ and HO$_x$-induced ozone destruction result only from chemical feedbacks through coupling to temperature or to Cl$_y$ reactions. The simulated PCO3$_{US}$ under near present day conditions (TS2000) is 63 DU. Net chemical loss is 48 DU day$^{-1}$, with the major loss being due to catalytic cycles involving NO$_x$ (39%), with smaller contributions from HO$_x$ (22%), halogens (20%) and O$_x$ (19%). The average chemical lifetime of ozone in the tropical upper stratosphere (calculated as the burden divided by net chemical loss) is 1.3 days. These results are consistent with previous studies (e.g. WMO, 1998; Grooß et al., 1999; Meul et al., 2014).

Comparison of TS2000$_{ODS}$ with TS2000 isolates the effects of future changes in ODSs on PCO3$_{US}$; as discussed in Section 3.2, we find that reductions in ESC from year 2000 values to projected values for year 2100 increase PCO3$_{US}$ abundances by 5 DU (8%). Table 2 shows that net chemical O$_x$ loss in TS2000$_{ODS}$ is reduced by 5% compared to TS2000, driven predominantly by large decreases in O$_x$ loss through catalytic cycles involving halogens, which are reduced by 63%. O$_x$ loss through reactions with HO$_x$, NO$_x$ and O$_x$ all increase, predominantly due to the increase in ozone concentrations, but also due to temperature changes, which are themselves a response to increases in ozone (e.g. Maycock, 2016). The upper stratosphere warms by ~2 K (Figure 4) when GHGs are held constant but ODS concentrations are reduced from year 2000 to year 2100 concentrations, consistent with the effect of increasing ozone concentrations on upper stratospheric temperatures as discussed by Maycock (2016). The reaction O+O$_3$ has a strong temperature dependence and becomes faster at higher temperatures, thereby further increasing O$_x$ loss in TS2000$_{ODS}$ relative to TS2000. Reactions involving HO$_x$ and NO$_x$ have weaker temperature dependencies and are coupled to Cl$_y$ concentrations through null cycles and the formation of reservoir species, and thus they show smaller increases.

In addition to the projected reduction in halogenated ODSs, the cooling of the stratosphere induced by increased GHG concentrations (mainly CO$_2$) will be a major driver of future PCO3$_{US}$ changes. Comparison of TS8.5 with TS2000 quantifies the impact of GHG changes alone on PCO3$_{US}$. As the chemical lifetime of O$_x$ is short in the upper stratosphere, transport changes are expected to have a relatively minimal effect on projected ozone trends. Instead, PCO3$_{US}$ changes between TS2000 and TS8.5 are driven by the response of reaction rates to the simulated temperature changes. The tropical upper stratosphere in TS8.5 is ~11 K cooler than in TS2000 (see Banerjee et al., 2016). This leads to a PCO3$_{US}$ increase of 12 DU

(21%), which is driven predominantly by a decrease in the reaction $O+O_3$, but also by a change in partitioning of $O_x$ due to the acceleration of the reaction $O+O_2+M \rightarrow O_3+M$ (Jonsson et al., 2004; Banerjee et al., 2016).

The relationship between $PCO3_{US}$ and upper stratospheric temperature for the transient and time-slice experiments is shown in Figure 4. From 1960 to 2000, temperatures and $PCO3_{US}$ both decrease. During this period the decrease in $PCO3_{US}$ is driven predominantly by increasing ODSs, as described above. Decreased ozone concentrations in turn reduce upper stratospheric heating, thereby reducing temperatures (e.g. Forster and Shine, 1997; Shine et al., 2003). From 2000 to 2100, as temperatures decrease further, mainly due to cooling from increasing $CO_2$ abundances, ozone concentrations increase, driven predominantly, as discussed above, by a reduced rate for the reaction of $O+O_3$ and decreased ODS concentrations. These increases in ozone offset part of the stratospheric cooling due to rising $CO_2$ concentrations (Maycock, 2016). The impact of temperature on $PCO3_{US}$ can be isolated by fitting lines through the sets of time-slice experiments with the same ODS loadings (i.e. TS2000, TS4.5 and TS8.5). We find that the relationship between $PCO3_{US}$ and upper stratospheric temperature is approximately $\frac{\Delta PCO_{US}}{\Delta T_{US}} = 1$ DU K$^{-1}$, which, when combined with decreasing ODS, drives the super-recovery of $PCO3_{US}$.

**4.2 Lower Stratosphere**

In comparison to the upper stratosphere, the chemical lifetime of $O_x$ in the tropical lower stratosphere is long (>1 month, see lower rows in Table 2), so dynamical processes play a much more important role in determining ozone abundances there. A strengthening of the BDC, which is projected to occur in the future in response to increases in GHGs (e.g. Shepherd and McLandress, 2011; Hardiman et al., 2014; Palmeiro et al., 2014), would therefore have a significant effect on tropical lower stratospheric ozone. We use the Transformed Eulerian Mean residual vertical velocity ($\overline{w}^*$; Andrews et al., 1987) at 70 hPa as a measure of the strength of the advective part of the BDC in the lower stratosphere. In the transient simulations, the annual and tropical (10°S-10°N) mean $\overline{w}^*$ at 70 hPa increases by around 40% from ~0.20 mm s$^{-1}$ in 1960 to ~0.28 mm s$^{-1}$ in 2100.

Consistent with the important role of the BDC in determining tropical lower stratospheric ozone abundances, there is a strong negative correlation (R = -0.76) between annual mean $PCO3_{LS}$ and $\overline{w}^*$ values at 70hPa (Figure 5a). By plotting $\overline{w_{70}^*}$ vs. CDE concentration as a function of time for the transient experiment (Figure 5b), and by comparing across the time-slice experiments with constant ODS loading (i.e. TS2000, TS4.5 and TS8.5), an approximate value for the acceleration of the BDC per unit increase in CDE can be calculated. From these experiments, a value of $\frac{\Delta \overline{w_{70}^*}}{\Delta CDE} \approx 2 \times 10^{-4}$ mm s$^{-1}$ ppmv$^{-1}$ is calculated. The strong negative relationship between $PCO3_{LS}$ and $\overline{w_{70}^*}$, and in concert the positive relationship between $\overline{w_{70}^*}$

and CDE concentration, combine to give a negative relationship between $PCO3_{LS}$ and CDE concentration, as shown in Figure 5c and quantified in Section 3.2.

The chemical effects of changing ODSs also impact on the modelled BDC strength. The TS2000 and $TS2000_{ODS}$ experiments are used to quantify this relationship as $\frac{\Delta \overline{w_{70}^*}}{\Delta ESC} \approx 5.4 \times 10^{-3}$ mm s$^{-1}$ ppbv$^{-1}$. This indicates that, per molecule, ODS
increases have a greater effect on the BDC than GHGs. Previous work using the UM-UKCA model has indicated that an acceleration in stratospheric circulation, particularly the lowermost branch of the BDC, is to be expected from increased springtime polar lower stratospheric ozone depletion and the resulting increase in meridional temperature gradients (Keeble et al., 2014; Braesicke et al., 2014). Our results also corroborate the findings of Polvani et al. (2017) who highlight the dominant impact of ODS on tropical lower stratospheric temperature and ozone through changes in tropical upwelling
between 1960-2000. Results from this study suggest that reduced future polar lower stratospheric ozone depletion following reduction in ODS concentrations will act to slow the BDC, partly offsetting the acceleration expected due to increased GHG concentrations.

The impact of ODS changes on the speed of the BDC, along with the temperature dependence of the ozone depleting chemistry and the influence of upper stratospheric ozone shielding on the lower stratosphere, result in a non-linear
dependence of the $PCO3_{LS}$ response to ODS on GHG loading, as was found in the upper stratosphere. In the time-slice experiments, the effect of the year 2000 to 2100 decrease in ODS on $PCO3_{LS}$ is: +7 DU for $TS2000_{ODS}$ - TS2000, +6 DU for $TS4.5_{ODS}$ - TS4.5 and +4 DU for $TS8.5_{ODS}$ − TS8.5 (compare circles and triangles of the same colour in Figures 5b and 5c). As described above, decreasing ODS concentrations lead to a deceleration of the BDC and an increase in $PCO3_{LS}$. However, as the stratosphere cools the increase in overhead ozone column reduces photolysis rates in the lower stratosphere, slowing
ozone production and acting to decrease $PCO3_{LS}$, as discussed above. Together these opposing mechanisms explain the difference in the $PCO3_{LS}$ response to ODSs changes under different GHG concentrations.

The combined influence of GHGs and ODS on the strength of tropical upwelling can largely explain the three distinct periods of behaviour in tropical $PCO3_{LS}$ seen in Figure 2b. Firstly, between 1960-2000, the partial column shows the largest rate of change as the effect of GHGs and ODS on tropical upwelling reinforce one another, both strengthening the tropical
upwelling and reducing $PCO3_{LS}$, while increasing stratospheric $Cl_y$ concentrations also enhance chemical ozone depletion. Secondly, between 2000-2040 increasing GHG concentrations lead to an acceleration of the BDC acting to reduce $PCO3_{LS}$ values while decreasing ODS concentrations slow the BDC and decrease chemical $O_x$ loss (Figure 5), and as such $PCO3_{LS}$ remains relatively constant during this time. Finally, between 2040-2100, by which time further changes in ozone and the BDC due to ODSs are reduced significantly, the effect of increasing GHGs on tropical upwelling dominates and $PCO3_{LS}$
values show a negative trend.

Finally, we note that in addition to changing to the strength of the BDC, increasing GHG concentrations also affect $PCO3_{LS}$ values by decreasing chemical production as a result of increased overhead column ozone (see Section 4.1). Table 2 shows how $O_x$ production in the lower stratosphere responds to changes in ODS and CDE concentrations. Compared to TS2000, lower stratospheric $O_x$ production in $TS2000_{ODS}$ and TS8.5 has decreased, consistent with the increased partial column abundances in the upper stratosphere in these simulations. Using this information we can calculate the response of lower stratospheric $O_x$ production to changes in upper stratospheric partial column abundance; we estimate that tropical lower stratospheric $O_x$ production will decrease by 0.1 DU day$^{-1}$ for each additional DU of ozone in the upper stratosphere.

## 4.3 Troposphere

The primary factors affecting future tropospheric ozone are likely to be changes in the emission of ozone precursors (CO, $CH_4$, $NO_x$ and VOCs) and changes in climate. Changes in climate can affect tropospheric ozone abundances in several ways, including changes in water vapour amounts, lightning $NO_x$ emissions ($LNO_x$) and stratosphere-troposphere exchange of ozone (STE) (e.g. Thompson et al., 1989; Young et al., 2013; Banerjee et al., 2016). Future ODS-driven stratospheric ozone recovery is also projected to increase tropospheric ozone abundances through STE (e.g. Zeng and Pyle, 2003; Banerjee et al., 2016). Here, we first use the time-slice simulations to deduce the role of climate change and ozone recovery on future tropospheric column ozone changes. Then, we discuss the likely drivers of the partial column evolution between 1960-2000 in the transient simulations, where changes in ozone precursors must also be considered.

Changes in the physical climate from increased concentrations of GHGs in the TS4.5 and TS8.5 experiments enhance tropical tropospheric column ozone by around 4 DU relative to TS2000. The increases are driven primarily by $LNO_x$, which increases by 2 and 4.7 Tg(N) yr$^{-1}$ under the RCP4.5 and RCP8.5 scenarios, respectively (Banerjee et al., 2014). In fact, a further sensitivity experiment in which the climate is allowed to change according to TS8.5, but $LNO_x$ values are kept fixed at TS2000 values (not otherwise discussed; see Banerjee et al., 2014) shows a 3 DU decrease in tropospheric column ozone. This reduction results from increases in tropospheric humidity under a warmer climate (e.g. Thompson et al., 1989). Thus, the increase in $LNO_x$ in TS8.5 contributes 7 DU to the increase in tropical $PCO3_T$.

A further increase in tropical $PCO3_T$ arises from the increase in the height of the tropopause under a warmer climate. In the transient simulations (which follow the RCP6.0 scenario), the ensemble mean annual mean tropopause height increases by 800 m from ~16.1 km to ~16.9 km between year 2000 and 2100. The impacts of increasing tropopause height on tropical $PCO3_T$ are calculated as the difference between the full $PCO3_T$ values calculated using a consistent tropopause height from the values calculated using a fixed year 2000 tropopause height of 16.1 km. This calculation indicates that the increase in tropopause height between 2000-2100 accounts for ~1.5 DU of the increase in tropical $PCO3_T$ in the transient experiment.

While reductions in ODS affect tropospheric ozone in the extratropics through STE (e.g. Banerjee et al., 2016), in the tropics, ODS have little impact on tropospheric ozone, with $PCO3_T$ increasing by <1 DU in the $TS2000_{ODS}$ experiment compared to TS2000 (see Figure 2c).

The time-slice sensitivity experiments indicate that the net effect of changes in the climate will be to increase tropical $PCO3_T$
in the transient simulations. However, in the transient simulations (run under all-forcings at RCP6.0) tropospheric ozone levels are also determined by the chemical effects of ozone precursors, including $CH_4$, CO and $NO_x$. The largest rate of change for tropical $PCO3_T$ occurs over the recent past (1960-2000) (see crosses in Figure 2c), during which time increases in anthropogenic $NO_x$ and $CH_4$ emissions have driven increases in tropospheric ozone production (e.g. Lamarque et al., 2010; Young et al., 2013).  After 2000, all the RCP scenarios project strong reductions in anthropogenic $NO_x$ and NMVOC
emissions (Meinshausen et al., 2011), which would in turn drive tropospheric ozone reductions. However, in the transient experiment tropospheric column ozone remains steady up to ~2040, partly due to the compensating effects of climate change, as suggested by the time-slice simulations, but also due to increasing tropospheric $CH_4$ concentrations (Young et al., 2013; Revell et al., 2015). The increase in tropical $PCO3_T$ up to 2060-2080 and its subsequent decline is consistent with the evolution of $CH_4$, which maximises around 2080 in the RCP6.0 scenario.

**5 Developing a simple model for predicting stratospheric column ozone change in the tropics**

Future projections of tropical TCO3 are strongly dependent on the assumed pathway for anthropogenic emissions, for which there is a great deal of uncertainty, particularly in relation to emissions of $CO_2$, $CH_4$ and $N_2O$.  CCMs are commonly used to assess possible future changes in ozone under a small number of well-defined scenarios, e.g. the RCP scenarios used in the IPCC Fifth Assessment Report (IPCC, 2013).  These emissions scenarios are neither forecasts nor policy recommendations,
but instead are chosen to represent a range of possible global socio-economic and technological pathways for the future.  In order to comprehensively quantify the response of the chemistry-climate system to such emissions scenarios, long, computationally expensive model simulations are required. However, simpler models can also be used to identify which processes dominate future trends and to explore the composition response to a wider range of emissions scenarios.

In Section 4, we quantified the impacts of halogen-catalysed ozone loss, changes in the strength of tropical upwelling and
upper stratospheric cooling induced by GHG changes (predominantly $CO_2$) on the tropical stratospheric ozone. Furthermore, the partial column ozone trends in the upper and lower stratosphere were found to be, to first order, linearly dependent on ESC and CDE concentrations (see Figures 3 and 5).  This conclusion was derived from the transient experiments adopting a single emissions scenario and multiple time-slice experiments based on 3 additional scenarios, and so is valid for a range of possible CDE and ESC concentrations.  In this section, we describe a simple, computationally inexpensive linearised model
that can be used to explore how tropical stratospheric column ozone may change under a much wider range of future ESC and CDE concentration pathways than are typically explored by comprehensive CCMs.  We emphasise that the experiments

described in this study do not allow us to distinguish the effects of other chemical species, such as $N_2O$ and $CH_4$, on stratospheric ozone, and thus the simple model does not attempt to include the effects of these species, which also vary substantially amongst RCP scenarios.

The simplest version of such a model has a linear dependence of tropical stratospheric column ozone (SCO3) on GHG concentrations (expressed in CDE) and ESC of the form:

$$SCO3_t = SCO3_{t_0} + \left( \frac{\Delta SCO3}{\Delta CDE} \cdot \left( CDE_t - CDE_{t_0} \right) \right) + \left( \frac{\Delta SCO3}{\Delta ESC} \cdot \left( ESC_t - ESC_{t_0} \right) \right),$$

where the subscripts $t_0$ and $t$ signify the reference year and the year the model is solving for, respectively. The constants $\frac{\Delta SCO3}{\Delta CDE}$ and $\frac{\Delta SCO3}{\Delta ESC}$, which represent the SCO3 change due to surface CDE and ESC perturbations, respectively, are calculated using the time-slice simulations which perturb ESC and GHGs separately. The parameter $\frac{\Delta SCO3}{\Delta ESC}$ is calculated by averaging the values obtained from the three pairs of simulations with different ODS loadings, but the same GHG concentrations, i.e. from the SCO3 differences between the green triangle and green circle in Figure 6 divided by the difference in surface ESC concentration between these runs. Similarly, the parameter $\frac{\Delta SCO3}{\Delta CDE}$ was calculated as the average of the linear fits through the three pairs of time-slice simulations with the same ODS loading, but different GHG concentrations, i.e. the green, blue and red circles in Figure 6. Using this method, values of $\frac{\Delta SCO3}{\Delta CDE}$ = -0.005 DU ppmv$^{-1}$ and $\frac{\Delta SCO3}{\Delta ESC}$ = -3.64 DU ppbv$^{-1}$ were obtained. The parameters for the simple model are therefore derived using the time-slice simulations and it can then be compared against the transient simulations in order to determine its ability to reproduce output from the same comprehensive CCM under a different scenario. These comparisons are shown in Figure 7 for the RCP4.5, 6.0 and 8.5 scenarios alongside annual mean stratospheric ozone column values from the transient UM-UKCA RCP6.0 simulations.

Projections of SCO3 made using the simple model following the RCP6.0 scenario for GHG and ODS (magenta line, Figure 7) can be compared with the fully-coupled transient simulations (grey lines, Figure 7). Overall the agreement between the simple model and the fully-coupled CCM is reasonable over the 140 year period considered. The simple model does capture the main features of modelled SCO3 values from the CCM, with rapid ozone loss in the late 20th century, a minimum around year 2000 and a gradual increase throughout the 21st century. Year 2100 SCO3 values are in good agreement between the simple model and CCM, with both indicating that tropical averaged SCO3 values will not return to their 1960 values despite reductions in halogenated ODS concentrations following the implementation of the Montreal Protocol. However, there are important quantitative differences between the simple model and fully coupled CCM results, which likely result from the neglect of additional important chemical controls on stratospheric ozone in the simple model (e.g. $N_2O$, $CH_4$). For example, the simple model overestimates the maximum extent of tropical SCO3 depletion occurring around year

2000, partly a result of the solar maximum in that year, and remain below the CCM values for the first half of the 21st century. Furthermore, the rate of increase in the later half of the 21st century is overestimated in the simple model compared to the CCM. This is likely due to the increased importance of $HO_x$ and $NO_x$ catalysed ozone destruction in the later part of the century associated with increases in $CH_4$ and $N_2O$ (e.g. Ravishankara et al., 2009; Fleming et al., 2011; WMO, 2014;

Butler et al., 2016), which are neglected in the simple model. In general though, there is good qualitative agreement between the simple model and CCM, which highlights the importance of GHG and ODS as major drivers of tropical SCO3 in the future.

As well as using the simple model to calculate SCO3 projections under the RCP6.0 scenario, additional emissions scenarios have also been investigated, and are also shown in Figure 7. These scenarios include the RCP4.5 and RCP8.5 scenarios

(green and yellow lines respectively), a scenario using time varying CDE concentrations following RCP6.0 with fixed 1960 values for ESC (light blue line) and a scenario using time-varying ESC concentrations with fixed 1960 values for CDE (dark blue line). All scenarios were initialised from 1960s SCO3 values in the transient experiment. The scenario from the simple model using fixed 1960 ESC values (light blue) highlights the projected decreases in tropical SCO3 resulting from GHG induced increases in the speed of the BDC, which is markedly different to the rest of the stratosphere where GHG induced

cooling leads to increased ozone mixing ratios. The results from the simple model indicate that by 2100 tropical SCO3 is lower following the RCP8.5 scenario, and higher following the RCP4.5 scenario, than the RCP6.0 scenario. This is because the reductions in lower stratospheric ozone from the acceleration of the BDC, which approximately scales with CDE (see Figure 5b), overwhelm any ozone increases in the upper stratosphere resulting from decreasing ESC concentrations and cooling of the upper stratosphere, as discussed in Section 4.2. These results from the simple model are in contrast to Eyring

et al. (2013a) who used output from CCMs that participated in CMIP5 and found that by 2100 SCO3 values are expected to be lowest under RCP6.0 and slightly higher under RCP8.5 (see their Figure 6b). This difference is partly due to not including the chemical effects of $N_2O$ and $CH_4$ in the simple model, as in the RCP8.5 scenario $CH_4$ levels at 2100 are more than double those in RCP4.5/6.0 and $N_2O$ values in 2100 are around 7% higher in RCP8.5 compared to RCP6.0 (Meinshausen et al., 2011). The differences between the end of 21st century SCO3 values in the simple model and the results

of Eyring et al. (2013a) may also reflect different sensitivities of UM-UKCA to radiative and chemical drivers compared to the CMIP5 multi-model ensemble. For example, the parameters $\frac{\Delta SCO3}{\Delta CDE}$ and $\frac{\Delta SCO3}{\Delta ESC}$ likely vary between different CCMs. Indeed, differences between these parameters in different CCMs would indicate varying sensitivites to GHG and ODS changes and may help in the identification of which processes have high uncertainty and should be explored in more detail.

Lastly, the parameters used in the simple model have been derived for a tropical band (10°N-10°S) but likely vary

substantially with latitude, so those calculated for this study could not be used to examine projections of extratropical SCO3 values. The aim of such a model is not to replace fully-coupled CCMs, but to provide a simple and computationally inexpensive way of exploring possible future SCO3 changes in the tropics. In this capacity, it appears to offer considerable

promise and could act as a valuable complementary approach to the 2D model studies which are currently used to investigate multiple scenarios (e.g. Fleming et al., 2011; WMO, 2014).

## 6 Conclusions

We have investigated the drivers of past and future changes in tropical averaged total column ozone using a number of model runs performed with two configurations of the UM-UKCA model. Four transient simulations following an RCP6.0 future GHG emissions scenario and WMO (2011) ODS recommendations were performed, with the longest of these simulations spanning the period 1960-2100. The transient runs were supplemented with 6 time-slice experiments run under a range of prescribed GHG and halogenated ODS loadings commensurate with either year 2000 or 2100 levels. Note that in the time-slice experiments only the chemical impacts of changes to ODS loadings and only the radiative impacts of GHG perturbations are considered, and so we focus on separating the contribution of these chemical and radiative drivers to future tropical ozone column changes. We do not consider explicitly in this study the chemical contributions to tropical column ozone trends of future $CH_4$ and $N_2O$ emissions. To aid in understanding the effects of the explored drivers on tropical column ozone changes, we analyse temporal trends in three partial ozone columns based on the following altitude ranges: the troposphere, the lower stratosphere (tropopause to 30km) and the upper stratosphere (30-48km). Ozone concentrations in each of these regions are governed by different processes and thus show distinct behaviours that combine to determine the overall evolution of total column ozone.

Future tropospheric ozone changes are driven by a number of processes, including changes to surface emissions of ozone precursors such as $CH_4$ and $NO_x$, increased $NO_x$ emissions from lightning associated with changes in convection, and changes to tropopause height. There is a high level of uncertainty associated with future emissions of ozone precursors, linked to uncertainties in anthropogenic emissions, biomass burning and land use changes. While the various RCP scenarios follow a range of future emissions scenarios for many key tropospheric ozone precursors, particularly $CH_4$, further work is required to explore the impact of changes to tropospheric ozone on TCO3 trends during the 21st century in order to understand to what extent changes in tropospheric ozone column offset decreases in the lower stratosphere. Of course the environmental benefits from reductions in tropospheric ozone as an air pollutant and GHG may considerably outweigh any gains increases in tropospheric ozone could have by balancing the effects of a decreased stratospheric ozone column on surface UV radiation.

The chemical effects of changes in ODSs and climatic changes due to GHGs drive changes to both the upper and lower stratospheric partial ozone columns. In the upper stratosphere, where the chemical lifetime of ozone is short (~ 1 day), projected future reductions in ODS concentrations and stratospheric cooling from increased GHG concentrations both lead to increased upper stratospheric partial column ozone by reducing halogen-catalysed destruction of ozone and slowing of the

temperature dependent ozone loss cycles, particularly those of the Chapman cycle, respectively. The combination of these two effects is expected to lead to super-recovery of upper stratospheric partial column values above their 1960s values.

Conversely in the lower stratosphere, where the chemical lifetime of ozone is typically >1 month, the partial column ozone values are predominantly controlled by changes to transport. Projected increases in GHGs lead to an acceleration of the

BDC, which is associated with increased transport of relatively ozone poor air masses into the tropical lower stratosphere, thereby decreasing ozone mixing ratios and the partial lower stratospheric column ozone. The magnitude of acceleration of the BDC is highly correlated with increasing GHG mixing ratios, and so the total effect of transport changes on tropical lower stratospheric ozone depends strongly on the future GHG emissions scenario. Future reductions in lower stratospheric ozone partial column values also result from decreased production of $O_x$ from photolysis of $O_2$ in the lower stratosphere due

to increased overhead ozone concentrations in the upper stratosphere. Analysis of the simulations presented here suggests lower stratospheric $O_x$ production will decrease by 0.1 DU day$^{-1}$ for each additional DU of ozone in the upper stratosphere.

The above points highlight that future projections of tropical stratospheric column ozone are the result of a complex interplay between drivers of ozone trends in the lower and upper stratosphere. The transient UM-UKCA simulations run under the RCP6.0 emissions scenario show that by the year 2100 stratospheric column ozone values are increased by 5 DU from the

minimum values around the year 2000. However, modelled stratospheric column values in the simulations never return to 1960s values despite declining stratospheric ODS loadings, due to the competing effects of changes in partial column ozone values in the lower and upper stratosphere.

Understanding the extent to which dynamically induced decreases in lower stratospheric partial column values counteract upper stratospheric super-recovery is key to making accurate projections of stratospheric column ozone, and requires

detailed modelling of both photochemical and dynamical processes under a range of future emissions scenarios. However, output produced by complex, fully-coupled CCMs can be used to create simple linear models which can be used to explore the stratospheric ozone column response to changing surface GHG and ODS concentrations. Simple linear models are computationally inexpensive and can be used to investigate a wide range of emission scenarios much more quickly than ensembles of fully-coupled CCMs. In this work we present a simple linearised model developed from the UM-UKCA

experiments to help investigate projections of stratospheric column ozone for a range of future emissions scenarios. The model includes parameters for the dependence of stratospheric column ozone on ESC and GHGs (expressed as Carbon Dioxide Equivalent) mixing ratios. The simple model was built using data from the time-slice UM-UKCA experiments and then its performance compared against the transient integrations. There is reasonable quantitative agreement between the simple model and the long-term behaviour of tropical stratospheric column ozone in the fully-coupled RCP6.0 CCM

simulations, confirming emissions of GHG and ODS to be key drivers of long-term future tropical stratospheric column ozone changes. However, there are quantitative differences between the results of the simple model for other RCP scenarios and previous multi-model results from CMIP5 (Eyring et al., 2013a). This is likely to be due to differences in $N_2O$ and $CH_4$

concentrations amongst RCP scenarios, which are neglected in the simple model, and may also be due to different models possessing different sensitivity parameters for stratospheric column ozone to changing ODS and GHG concentrations.

In summary, while fully-coupled CCM simulations are required to precisely quantify changes in, and identify the processes responsible for, future atmospheric composition changes, simple models can provide a complementary approach for investigating a broad range of potential emissions scenarios. Furthermore, it is hoped that the model presented here can be further developed to include more parameters (e.g. $N_2O$, $CH_4$) by performing more integrations, and also to more accurately constrain the terms of the simple model by using integrations from more CCMs. This would also allow for a better assessment of the uncertainty of each of the terms used in the simple model.

## Availability of data

Data from the transient simulations are available as part of the CCMI initiative through BADC: https://blogs.reading.ac.uk/ccmi/badc-data-access/. All further data are available upon request.

## Acknowledgements

The research leading to these results has received funding from the European Community's Seventh Framework Programme (FP7/2007 - 2013) under grant agreement n° 603557 (StratoClim), the European Research Council through the ACCI project (project number: 267760) and the Natural Environment Research Council through the CAST project (NE/ I030054/1) and ACM's Independent Research Fellowship (NE/M018199/1). We thank NCAS-CMS for modelling support. Model integrations have been performed using the ARCHER UK National Supercomputing Service and MONSooN system, a collaborative facility supplied under the Joint Weather and Climate Research Programme, which is a strategic partnership between the UK Met Office and the NERC.

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

| Simulation name | Climate (SST, sea ice, GHG) | ODS ($Cl_y$, $Br_y$) |
| --- | --- | --- |
| TS2000 | 2000 | 2000 |
| TS2000$_{ODS}$ | 2000 | 2100 (RCP4.5) |
| TS4.5 | 2100 (RCP4.5) | 2000 |
| TS4.5$_{ODS}$ | 2100 (RCP4.5) | 2100 (RCP4.5) |
| TS8.5 | 2100 (RCP8.5) | 2000 |
| TS8.5$_{ODS}$ | 2100 (RCP8.5) | 2100 (RCP4.5) |

**Table 1.** Simulation names and corresponding climate (including radiative impacts of GHGs, SSTs and sea ice) and ODS loadings. Note that changes in halogenated ODSs are imposed only on the chemistry scheme while changes in GHGs ($CO_2$, $CH_4$, $N_2O$ and CFCs) are imposed only on the radiation scheme. RCP scenario used for future GHG and ODS 5 concentrations given in parentheses.

| | Integration | PCO3 | Lifetime | Production | Loss | Halogens | $HO_x$ | $NO_x$ | $O_x$ |
|---|---|---|---|---|---|---|---|---|---|
| $PCO3_{US}$ | TS2000 | 63 DU | 1 day | 48 DU day$^{-1}$ | 48 DU day$^{-1}$ | 9 DU day$^{-1}$ | 11 DU day$^{-1}$ | 19 DU day$^{-1}$ | 9 DU day$^{-1}$ |
| | TS2000$_{ODS}$ | +8% | +13% | -5% | -5% | -63% | +9% | +4.5% | +19% |
| | TS4.5 | +5% | +8% | -3% | -3% | +2% | +2% | -5% | -7% |
| | TS8.5 | +19% | +27% | -6% | -6% | +4% | +11% | -13% | -21% |
| $PCO3_{LS}$ | TS2000 | 179 DU | 34 days | 7 DU day$^{-1}$ | 5 DU day$^{-1}$ | 1 DU day$^{-1}$ | 2 DU day$^{-1}$ | 2 DU day$^{-1}$ | 1 DU day$^{-1}$ |
| | TS2000$_{ODS}$ | +4% | +21% | -10% | -14% | -65% | -6% | +3% | +7% |
| | TS4.5 | -3% | +8% | -8% | -10% | -5% | -8% | -10% | -8% |
| | TS8.5 | -7% | +23% | -16% | -25% | -17% | -15% | -33% | -37% |

**Table 2.** Partial column ozone values (DU), average ozone lifetime (days), net chemical production and loss and absolute contribution of halogen, $HO_x$, $NO_x$ and $O_x$ ozone destroying cycles (DU day$^{-1}$) in the upper and lower stratopsheric for the TS2000 integration. Percentage change for the TS2000$_{ODS}$, TS4.5 and TS8.5 simulations relative the to TS2000 simulation.

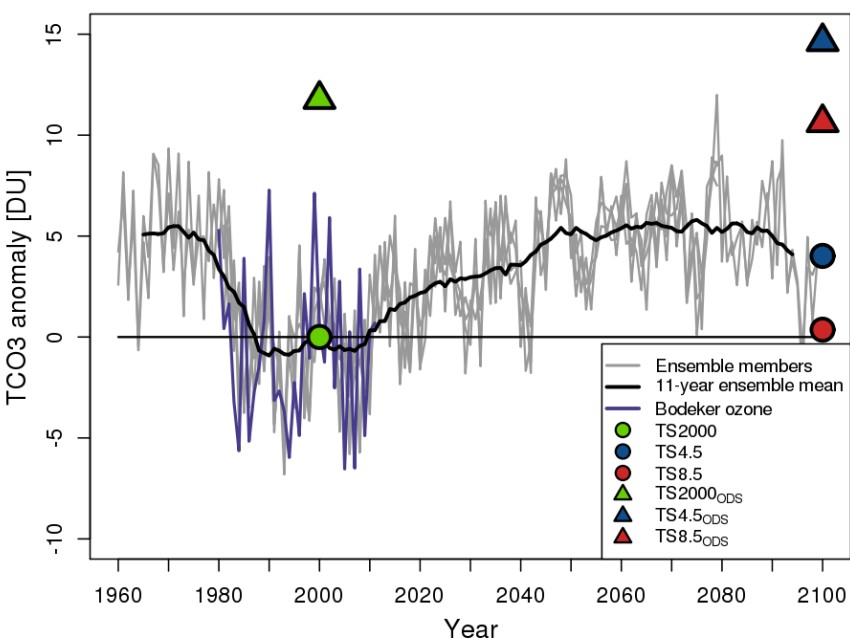

**Figure 1.** Total column ozone anomalies (in DU) relative to the year 2000±5 mean, averaged over 10°S-10°N for the four transient UM-UKCA experiments following the RCP6.0 future emissions scenario (grey lines), and the ensemble mean 11-year running mean (black line). Coloured circles and triangles represent tropical total column ozone in the time-slice experiments, as given in the figure legend. The purple line shows tropical averaged total column ozone values from v2.8 of the Bodeker dataset (Bodeker et al., 2005).

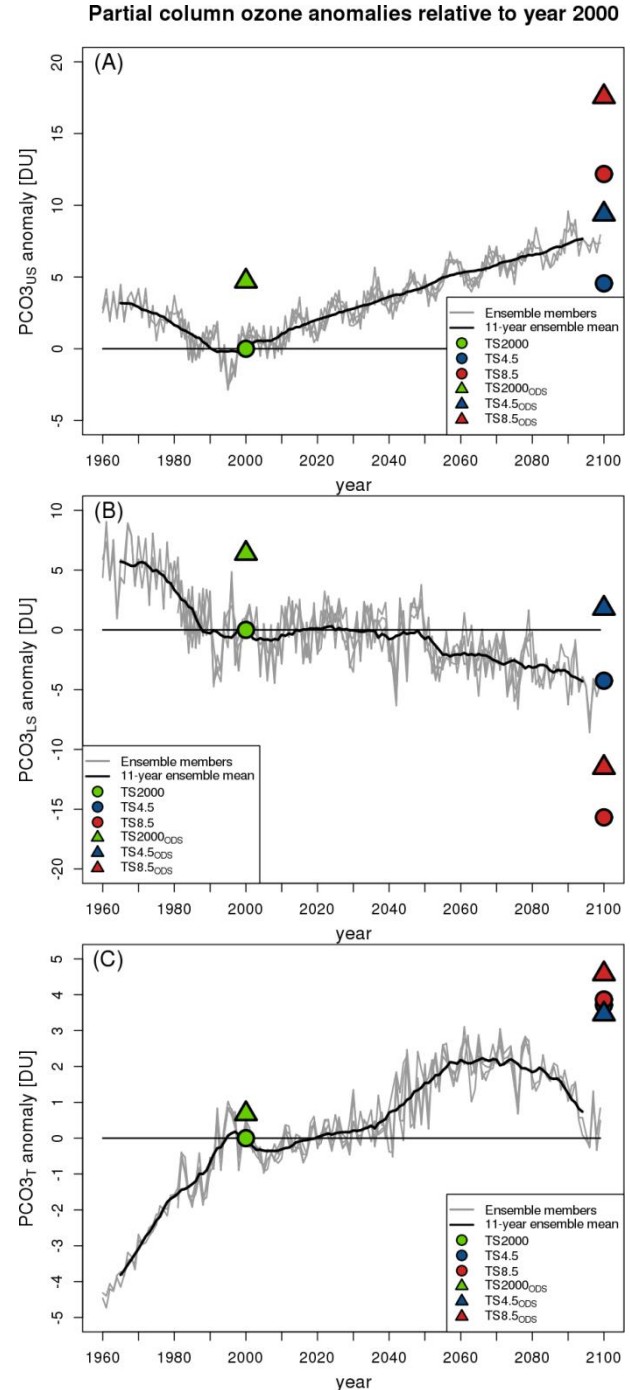

**Figure 2.** As for Figure 1, but for partial columns for (a) the upper stratosphere (30-48 km), (b) lower stratosphere (tropopause-30 km) and (c) troposphere.

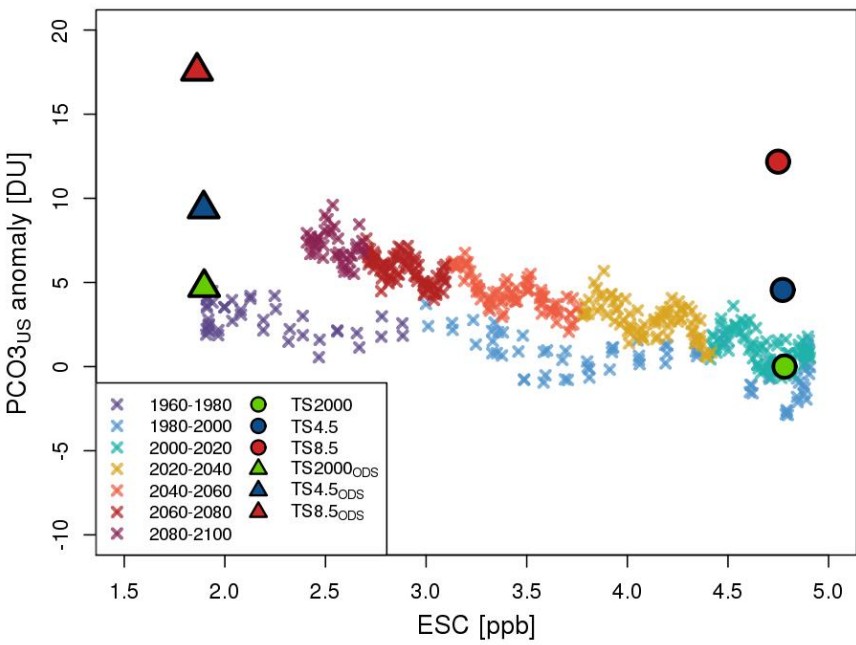

**Figure 3.** Scatterplot of annual mean upper stratospheric partial column ozone anomalies relative to the year 2000±5 mean (in DU) vs. 45km ESC (in ppb) for the transient simulation (crosses) and time-slice experiments (circles and triangles). Results from the transient simulations have been coloured in 20 year sections.

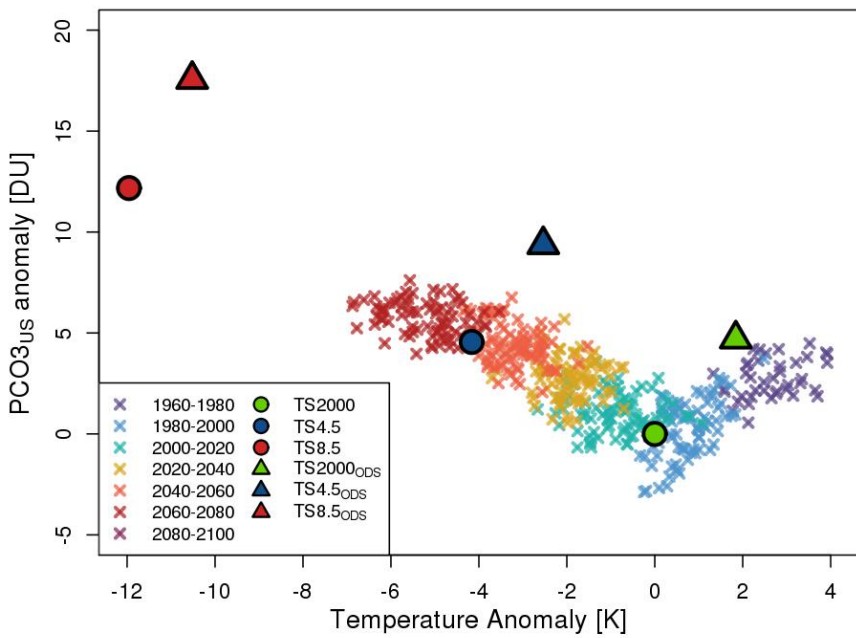

**Figure 4.** Scatterplot of annual mean upper stratospheric partial column ozone anomalies relative to the year 2000±5 mean (in DU) vs. 45km temperature (in K) for the transient simulation (crosses) and time-slice experiments (circles and triangles). Results from the transient simulations have been coloured in 20 year sections.

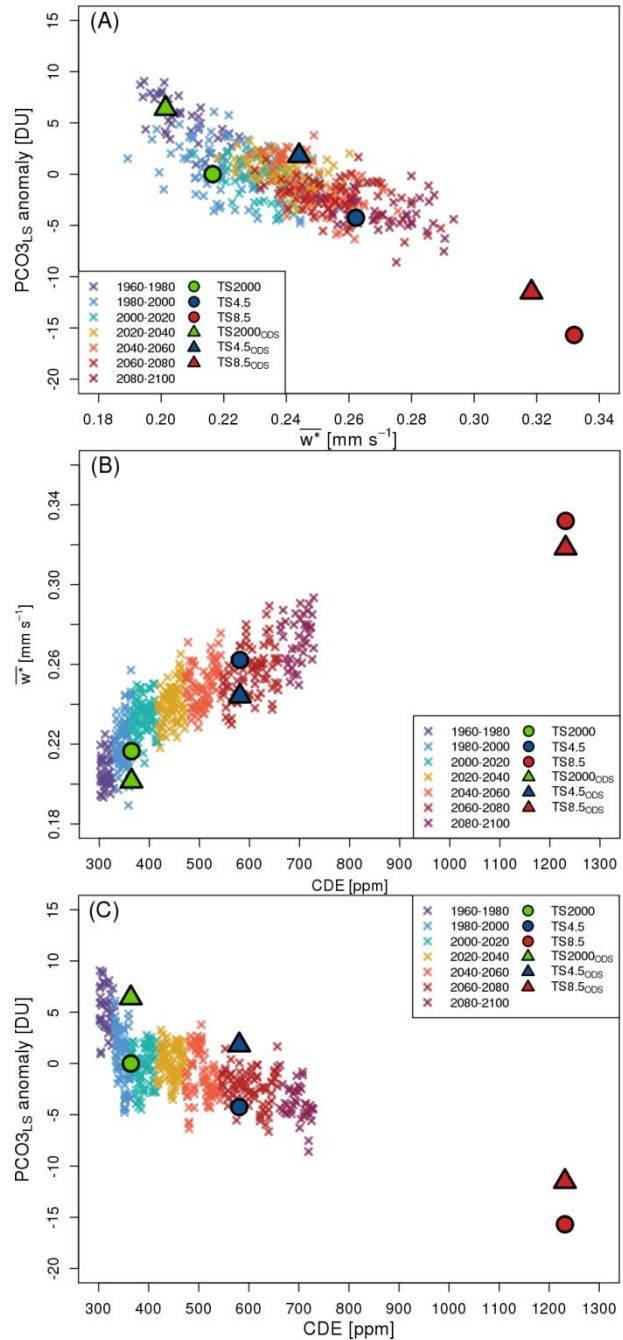

**Figure 5.** Scatter plot of (a) lower stratospheric partial column ozone anomalies relative to the year 2000±5 mean (in DU) vs. 70hPa $\overline{w}^*$ (in mm/s), (b) 70hPa $\overline{w}^*$ vs. CDE mixing ratio (in ppmv) and (c) lower stratospheric partial column ozone

anomalies vs. CDE mixing ratio for the transient simulations (crosses) and time-slice experiments (circles and triangles). Results from the transient simulations have been coloured in 20 year sections.

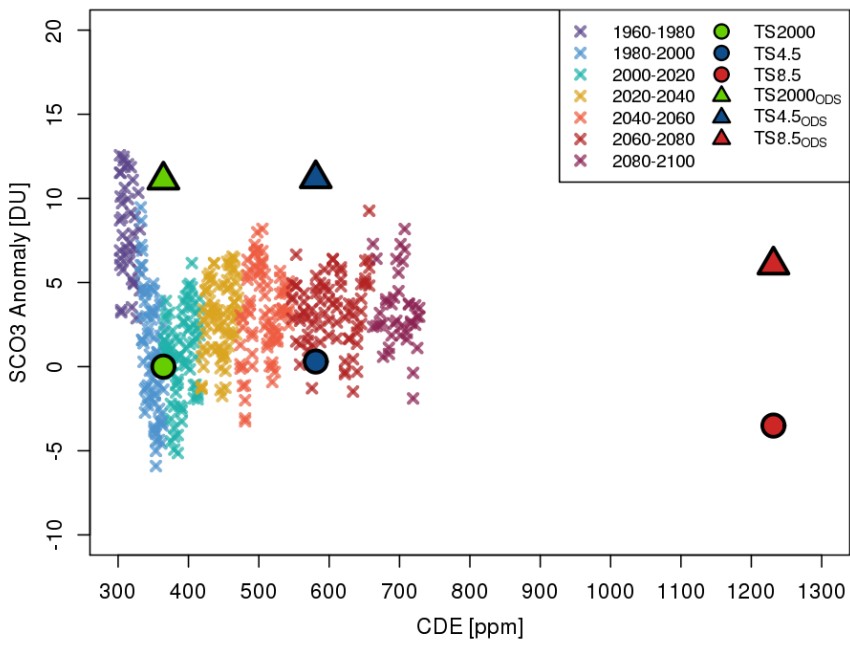

**Figure 6.** Scatter plot of stratospheric column ozone anomalies relative to the year 2000±5 mean (in DU) vs CDE mixing ratio (in ppmv) for the transient simulation (crosses) and time-slice experiments (circles and triangles). Results from the transient simulations have been coloured in 20 year sections.

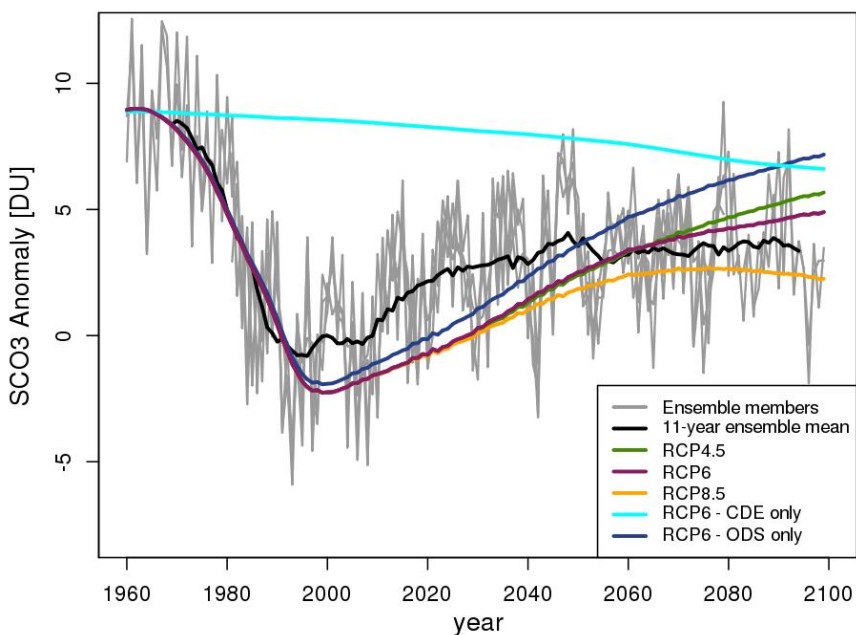

**Figure 7.** Annual mean stratospheric column ozone anomalies relative to the year 2000±5 mean (in DU) as modelled by the transient simulations (grey lines), with the ensemble mean 11-year running mean also plotted (black line). Results obtained using the simple model are shown for a range of emissions scenarios, initialised to 1960 values taken from the transient run.