# Peer review of "Diagnosing the radiative and chemical contributions to future changes in tropical column ozone with the UM-UKCA chemistryclimate model"

_Atmospheric Chemistry and Physics, 2017_

## Referee Comment (RC1) · Anonymous Referee #1 · 26 Apr 2017

General comments

Keeble et al. examine tropical ozone trends between 1960-2100 in an ensemble of chemistry-climate model simulations following RCP 6.0. They examine trends in the upper stratosphere, lower stratosphere and troposphere, and use a set of sensitivity simulations to quantify the chemical effects of CFCs, and the radiative effects of greenhouse gases (CO2 + N2O + CH4 + CFCs). They have laid the foundation for a thorough analysis of projected tropical ozone trends, which will be of interest for the stratospheric ozone community, however I do have a number of issues with the paper in its present form that I think should be addressed before the paper is published in ACP.

[Figure]

The authors do not include the chemical effects of CH4 and N2O in their sensitivity simulations. As the authors themselves note (P3L14-16): "the atmospheric concentration of these species, and by extension future concentrations of HOx and NOx radicals, is therefore highly sensitive to assumptions made about their future emissions." I would have thought this a good reason to include them in the analysis, particularly as CH4 and N2O are not currently regulated, unlike the CFCs. I also do not agree with statements such as (p.13): "...we showed that future changes in tropical stratospheric column ozone are driven primarily by changes in: (i) the halogen-catalysed loss; (ii) the strength of tropical upwelling; and (iii) the upper stratospheric cooling induced by GHGs (mainly CO2)." You did not look at changes in N2O and CH4 chemistry , so how can you say that they are not important drivers? Or, that "the changes in HOx and NOx chemistry resulting from future changes in CH4 emissions would appear to be of second order on the timescales considered" (P13L27-28). A number of studies show that stratospheric ozone is controlled by CO2, CH4 and N2O in the second half of the 21st century (see e.g. Butler et al. 2016 and references therein), and I think the authors need to address this. N2O is the most important ODS currently emitted (Ravishankara et al., 2009), and while reductions in CFCs and increases in CO2 will have a major effect on ozone this century, I am skeptical that N2O can be considered of secondary importance, especially since its chemical effects were not included in this analysis.

I would also like to see a fuller discussion of how the authors' results compare with existing chemistry-climate model studies. For example, they could be compared with the sensitivity studies of Butler et al., 2016; Eyring et al., 2010; Fleming et al., 2011; Oman et al., 2010 and Revell et al., 2012; full citations are given below. Do the authors' results confirm results from existing studies? Do they show something new?

As well as comparing the results with other model studies, I think the authors should compare their RCP 6.0 simulation with observations where available, to show how well their CCM performs in the tropics.

I am not convinced that the simple model discussed in Section 5 is reliable. It shows

(Fig. 7) that stratospheric ozone abundances at the end of the 21st century are higher in RCP 4.5 than they are in RCP 6.0, which is higher again than in RCP 8.5. This is in direct contrast to results from existing CCMs, which show that ozone is highest in RCP 8.5 > RCP 6.0 > RCP 4.5 (see e.g. Fig. 2-23 from Chapter 2 of the WMO 2014 Ozone Assessment). And why does ozone decrease over time when ODSs are held fixed – surely GHG-induced stratospheric cooling should cause ozone to increase? C.f. e.g. Fig. 6 from Fleming et al. (2011).

The figures are generally well presented. I do have some ideas for splitting them up and recombining the various subfigures to improve the flow of the discussion (noted later on). The tables contain a few errors, which I have also noted later on in this review.

Specific comments

- The authors repeatedly refer to ozone recovery and "super-recovery." I understand what they are referring to, however the terminology is not correct. Ozone is projected to increase through the 21st century because (i) CFCs decrease; (ii) GHG-induced stratospheric cooling (mostly by $CO_2$) increases. Any ozone increase induced by (ii) is not a "recovery," because it was CFCs that caused late 20th century ozone depletion in the first place. I would prefer that such statements surrounding recoveries and super-recoveries are worded more carefully.

- Reactive chlorine is referred to as $Cl_y$ and $ClO_x$. It would improve readability if one term was used consistently.

- Be careful when referring to ODS-driven ozone loss. Here you refer to ODSs (in your timeslice simulations) as $Cl_y + Br_y$ containing species, and do not include $N_2O$, which is also an ODS.

- The discussion of partial column ozone differences (3.2) is difficult to interpret since the drivers of ozone change are given only in the following section. The discussion of

drivers of ozone change needs to come sooner. I suggest splitting up figure 2, and combining fig. 2a with figs. 3 and 4; combining fig. 2b with fig. 5; and combining fig. 2c with zonally-resolved plots (discussed later in this review). Then the partial column differences and their drivers in each region of the atmosphere can be discussed sequentially.

- In the introduction, you discuss the benefits of the stratospheric ozone layer for human health, however a discussion of the harmful effects on tropospheric ozone (as an air pollutant and GHG, and its negative effects on visibility and crop damage) is missing.

- P2L1-2: At first this reads like a contradiction. The authors need to explain that in the tropics there is a small stratospheric ozone column with a high ozone concentration, and a large tropospheric ozone column with a low ozone concentration, because of the higher tropopause.

- P2L3-4: You might also want to mention projected changes in tropospheric ozone precursors from developing countries.

- P2L10-11: note the time period these studies looked at: they show that tropical TCO3 might not reach pre-1980s values by the end of the 21st century.

- P2L18: See e.g. Solomon et al., 2016.

- P3L28: While emissions to date indicate that RCP 8.5 is "business-as-usual" at present, I am uncomfortable referring to RCP 8.5 in this way since the methane concentrations by the end of the 21st century are so extreme.

- P4L16: "WCRP/SPARC" -> "IGAC/SPARC" also the reference Eyring et al. (2013) should be changed to Morgenstern et al., 2017. You could also include a sentence describing what CCMI is.

- P4L17: Was the chemistry scheme UKCA or CheS+? How are they different? Please provide more details here.

- P4L23-24: How were the initial atmospheric conditions perturbed for each ensemble run?

- Table 1 has some errors. I think TS4.5_ODS is supposed to read: climate = 2100 (RCP 4.5) and for TS8.5 climate = 2100 (RCP 8.5).

- The ODS scenarios developed for the RCPs are all rather similar and similar to the WMO A1 scenario for halocarbons, is this correct? You may want to include some detail here and thus justify why you use Year 2100 ODSs from RCP 4.5 in you TS8.5_ODS simulation.

- It would be helpful in Table 1 to note that changes in ODSs (Cly and Bry species but not NOx) are imposed only on the chemistry scheme while changes in GHGs (incl. CFCs) are imposed only on the radiation scheme.

- For experiment TS2000 do GHGs (i.e. CH4 and N2O) influence chemistry? I think so as this is your "base" run and the other five timeslice experiments are the perturbation experiments, is that correct?

- P5L18: Stating that a full description of the simulations is available in Banerjee et al. (2014) is not very helpful as they use a different nomenclature. Please include all relevant details here.

Figure 1:

- I would like to see some evaluation and discussion of how your CCM performs compared to observations; maybe by plotting observations for when they are available on Figure 1.

- I am surprised that tropical total column ozone increases by so much in the mid-21st century (Fig. 1), and would like to see more discussion on this, as it is somewhat at odds with the existing literature (see e.g. Fig.2-23 of the WMO 2014 Ozone Assessment, Chapter 2; Fig. 6 of Eyring et al. (2013)). Is the upper stratospheric cooling in the model excessive? Or is too much ozone produced in the troposphere, for example?

- I am missing a discussion of why TCO3 in the TS2000_ods experiment is so high (higher than in 1960 in the RCP 6.0 simulation). I think this could be because there is very little Cly-induced ozone loss, together with a strong radiative effect from GHGs, which cools the upper stratosphere and thus increases ozone – please discuss this.

- P6L23-24: Please be more explicit here. Ozone-destroying chlorine chemistry is temperature-dependent, therefore slows in a colder stratosphere, therefore ozone increases.

- P7L3-4: Was there a particular reason that you chose 30 km to differentiate between the upper and lower stratosphere? Please also state the pressure level.

- P7L8-11: State why ozone increases, i.e. the GHG-induced stratospheric cooling effect.

- P712-13: But as already stated by the authors, the effect of Cly forcing is non-linear and dependent on the climate scenario. So what does it mean to say that a 5 DU increase in ozone can be attributed to Cly over the 21st century, given that you are looking at a year 2000 climate? I think you're getting at that if ODS concentrations in 2000 were equal to the year 2100 values, we would expect ozone to be 5 DU higher, right?

- P7L15: Please be more explicit here about what the Maycock 2016 paper shows – it looks as though you cite it to back up the statement that stratospheric cooling is GHG scenario dependent, but this has been known for a long time.

- P7L22-23: Are you referring to the difference between the blue circle and triangle, and the difference between the red circle and triangle? It is hard to read from the figure, but looks like it is ∼5 DU for each. That is indeed interesting – it implies that in the upper stratosphere, the climate scenario has little effect on Cly-induced ozone destruction? Why would that be?

- P8L2: "compare red/blue circles with green circle in Figure 2b" – this sort of statement

is useful in interpreting the figures, and I encourage the authors to use more of them.

- P8L6-8: this sentence is confusing; please reword it. Namely, what are you comparing to the upper stratosphere?

- P8L12-14: Why? Evolution of ozone precursor emissions in RCP 6.0 due to countries cleaning up their air quality?

- P8L15: How do your results compare with the ACCMIP models? (Young et al., 2013).

- P816-17: You might want to state that this is expected because ODSs are photolysed in the stratosphere, not the troposphere.

- Table 2: how are the contributions to ozone destruction calculated?

- Table 2: You show NOx and HOx-induced ozone destruction, although chemical changes in N2O and CH4 were not included in simulations TS2000_ODS, TS4.5 and TS8.5. . . I think you should state this in the table caption to make it clear that any changes in their rates are radiative effects or buffering by Cly.

- P9L17: State how much of a reduction in EESC induces an increase in PCO3_US by 5 DU.

- P9L19-20: Ox loss through reactions with Ox? Rather the Chapman cycles?

- P9L22: The upper stratosphere warms when GHGs are held constant but Cly is decreased from 2000 to 2100 concentrations. Please clarify this.

- P9L24: But as well as temperature effects, HOx and NOx cycles will also be buffered by interactions with Cly. This should also be discussed.

- P10L1-18: As mentioned earlier, it would be great if the discussion of ozone drivers came earlier.

- Figure 5c is not discussed in the text.

- P11L12-14: How are non-linearities accounted for here?

- P11L16-17: CO is also an important ozone precursor.

- Figure 2c: In the tropical troposphere, different chemistry regimes are at play, and a lot of information can be lost through zonal averaging. For example, in the tropical Western Pacific region ozone loss via the H2O + O(1D) reaction is very important where solar actinic fluxes and humidity are high. However in other regions, ozone production can dominate due to anthropogenic emissions of ozone precursors (biomass burning etc). I think it would be interesting to somehow resolve figure 2c zonally, and discuss a bit more the chemical changes happening there.

- P12L8-9: State where this is shown (Fig. 2c).

- P12L11: I would argue that ozone precursors are a major consideration, rather than an additional consideration... I think you could look at their effects here too, as from Banerjee et al. (2016) I understand you have simulations available where climate and ozone precursor emissions are perturbed separately and together?

- How were ozone precursor emissions prescribed in your timeslice simulations? The same as RCP 6.0?

- P12L17: Also compare with the ACCMIP models in Young et al. (2013).

- P14L6-7: was CDE fixed or CO2? In the text you say that CDE was fixed, but in the legend on Fig. 7 it says that CO2 was fixed. Please use consistent terminology. I think too that the caption for Fig. 7 should provide a description of the experiments shown.

- Figure 7: Why does the simple model overestimate ozone loss between ~1990-2070?

- Discussion of fig. 7: Non-linearities are not discussed; (Meul et al., 2015) may provide helpful background information here.

- P15L7: you are talking in terms of the total column, right? Again, I am missing a discussion of the role of tropospheric ozone as an air pollutant – even if lower stratospheric ozone losses are balanced in the total column by tropospheric increases, the

result is not great for life in the biosphere because of reduced stratospheric ozone shielding the biosphere from UV-B radiation, and increased tropospheric ozone acting as an air pollutant and GHG.

- P15L8-10: Again, I disagree since these were the only factors you looked at, so you cannot discount other factors.

- P15L28-30: This was not discussed earlier, please include this discussion in the results section.

- Please state where your data are available from.

Technical corrections

- P1L18 "significant differences to" -> "significant differences in"

- P2L6: Montreal Protocol and its subsequent amendments -> Montreal Protocol and its subsequent Adjustments and Amendments

- P2L21: "over the course of the 21st century perturb" -> "over the course of the 21st century are expected to perturb"

- P2L23-24: CFCs are source gases for Cly, N2O is a source gas for NOx and CH4 is a source gas for HOx. Please phrase this more carefully.

- P2L26: "increases to the rate constant" -> "increases in the rate constant"

- P2L27: "decreases to the rate constant" – as above.

- P3L6: define Cly and NOy.

- P3L24 onwards: there is no need to refer to "RCP emissions scenarios" or "RCP scenarios." Calling them RCPs is sufficient.

- P3L28: "rise" -> "increase"

- P5L3: "integration given" -> "integration are given"

- P6L16: "discussed in" -> "discussed by"

- P7L14: "century is dependent" -> "century are dependent"

- P8L5-6: units are in italics.

- P9L3: "62 DU" – it says 63 DU in Table 2.

- P11L18: NOx: fix subscript.

- P11L30: "increase in LNOx at RCP 8.5" -> "increase in LNOx in RCP 8.5"

- P12L15: Meinhausen -> Meinshausen

- P13L26 "emissions of GHGs" -> "the radiative effects of GHG emissions"

- P13L28: dynamic -> dynamical

- P15L3: troposphere height -> tropopause height

References

Butler, A.H., Daniel, J.S., Portmann, R.W., Ravishankara, A.R., Young, P.J., Fahey, D.W., Rosenlof, K.H., 2016. Diverse policy implications for future ozone and surface UV in a changing climate. Environmental Research Letters 11.

Eyring, V., Cionni, I., Bodeker, G.E., Charlton-Perez, A.J., Kinnison, D.E., Scinocca, J.F., Waugh, D.W., Akiyoshi, H., Bekki, S., Chipperfield, M.P., Dameris, M., Dhomse, S., Frith, S.M., Garny, H., Gettelman, A., Kubin, A., Langematz, U., Mancini, E., Marchand, M., Nakamura, T., Oman, L.D., Pawson, S., Pitari, G., Plummer, D.A., Rozanov, E., Shepherd, T.G., Shibata, K., Tian, W., Braesicke, P., Hardiman, S.C., Lamarque, J.F., Morgenstern, O., Pyle, J.A., Smale, D., Yamashita, Y., 2010. Multi-model assessment of stratospheric ozone return dates and ozone recovery in CCMVal-2 models. Atmospheric Chemistry and Physics 10, 9451–9472.

Fleming, E.L., Jackman, C.H., Stolarski, R.S., Douglass, A.R., 2011. A model study of the impact of source gas changes on the stratosphere for 1850–2100. Atmospheric

[Figure]

Chemistry and Physics 11, 8515–8541. Meul, S., Oberländer-hayn, S., Abalichin, J., Langematz, U., 2015. Nonlinear response of modelled stratospheric ozone to changes in greenhouse gases and ozone depleting substances in the recent past. Atmospheric Chemistry and Physics 15, 6897-6911.

Morgenstern, O., Hegglin, M.I., Rozanov, E., O'Connor, F.M., Abraham, N.L., Akiyoshi, H., Archibald, A.T., Bekki, S., Butchart, N., Chipperfield, M.P., Deushi, M., Dhomse, S.S., Garcia, R.R., Hardiman, S.C., Horowitz, L.W., Jöckel, P., Josse, B., Kinnison, D., Lin, M., Mancini, E., Manyin, M.E., Marchand, M., Marécal, V., Michou, M., Oman, L.D., Pitari, G., Plummer, D.A., Revell, L.E., Saint-Martin, D., Schofield, R., Stenke, A., Stone, K., Sudo, K., Tanaka, T.Y., Tilmes, S., Yamashita, Y., Yoshida, K., Zeng, G., 2017. Review of the global models used within phase 1 of the Chemistry–Climate Model Initiative (CCMI). Geosci. Model Dev. 10, 639-671.

Oman, L.D., Waugh, D.W., Kawa, S.R., Stolarski, R.S., Douglass, A.R., Newman, P.A., 2010. Mechanisms and feedback causing changes in upper stratospheric ozone in the 21st century. Journal of Geophysical Research 115, D05303, doi:05310.01029/02009JD012397.

Ravishankara, A.R., Daniel, J.S., Portmann, R.W., 2009. Nitrous Oxide (N2O): The Dominant Ozone-Depleting Substance Emitted in the 21st Century. Science 326, 123-125.

Revell, L.E., Bodeker, G.E., Smale, D., Lehmann, R., Huck, P.E., Williamson, B.E., Rozanov, E., Struthers, H., 2012. The effectiveness of N2O in depleting stratospheric ozone. Geophysical Research Letters 39, doi:10.1029/2012GL052143.

Solomon, S., Kinnison, D., Garcia, R.R., Bandoro, J., Mills, M., Wilka, C., Neely III, R.R., Schmidt, A., Barnes, J.E., Vernier, J., Höpfner, M., 2016. Monsoon circulations and tropical heterogeneous chlorine chemistry in the stratosphere. Geophysical Research Letters 43, 624-633.

Young, P.J., Archibald, A.T., Bowman, K.W., Lamarque, J.F., Naik, V., Stevenson, D.S., Tilmes, S., Voulgarakis, A., Wild, O., Bergmann, D., Cameron-Smith, P., Cionni, I., Collins, W.J., Dalsøren, S.B., Doherty, R.M., Eyring, V., Faluvegi, G., Horowitz, L.W., Josse, B., Lee, Y.H., MacKenzie, I.A., Nagashima, T., Plummer, D.A., Righi, M., Rumbold, S.T., Skeie, R.B., Shindell, D.T., Strode, S.A., Sudo, K., Szopa, S., Zeng, G., 2013. Pre-industrial to end 21st century projections of tropospheric ozone from the Atmospheric Chemistry and Climate Model Intercomparison Project (ACCMIP). Atmospheric Chemistry and Physics 13, 2063-2090.
* * *

---

## Referee Comment (RC2) · Anonymous Referee #2 · 3 May 2017

The manuscript presents an investigation of the model-projected evolution of ozone in the tropics (10S to 10N) over the period 1960 to 2100. A transient simulation covering the whole period and following the specified reference scenario for the Chemistry-Climate Model Initiative (CCMI) model intercomparison project, the REF-C2 simulation, is augmented with time-slice simulations for year 2000 and 2100 conditions run under different levels of ozone depleting substances. The influence of changing greenhouse gases (GHGs) and ozone depleting substances (ODSs) on tropical total column ozone are investigated by splitting the total column into upper stratospheric, lower stratospheric and tropospheric components. Linear functions of the change in column ozone due to the effects of GHGs and ODSs are derived and these functions are used to

reproduce the evolution of stratospheric ozone column in the full model simulation.

Of significance, the results add to a number of recent papers that underline the importance of ODS-driven changes in ozone on tropical upwelling in the lower stratosphere. The core of the methodology and results presented in the manuscript are, in my opinion, solid. My one significant concern is the way in which the effects of methane and nitrous oxide are treated. On page 13, Lines 1-4, the authors state:

'In Section 4 we showed that future changes in tropical stratospheric column ozone are driven primarily by changes in: (i) the halogen-catalysed loss; (ii) the strength of tropical upwelling; and (iii) the upper stratospheric cooling induced by GHGs (mainly $CO_2$).'

I would argue that the authors have not, in fact, shown this in general. The inferred causes of changes in partial column ozone are derived from the set of timeslice experiments that only varied GHGs and ODSs. That these are then the only two factors that were found to be responsible for changes in ozone should naturally follow. On Page 5, Lines 9-11 the authors state 'In this study we consider the radiative impact of a large number of GHG species ($CO_2$, $CH_4$, $N_2O$, CFCs) and assume that the dominant driver of chemical changes is changes to ODS loadings. In this way, the chemical impact of changing $N_2O$ and $CH_4$ emissions is not considered here.' Since the effects of changing $N_2O$ and $CH_4$ are not considered it seems difficult to justify the conclusion (Page 16, Lines 21-23) that 'Results from the simple model indicate stratospheric column ozone changes resulting from future $CH_4$ and $N_2O$ emissions are of second order on the timescales considered here.'

While the parameterization of stratospheric column that is derived here is able to reproduce fairly well the evolution of stratospheric column in the transient simulation, the variation of methane is fairly small in RCP6. The parameterized stratospheric column also significantly overestimates the trend from 2020 to 2100, where the full model shows almost no change while the parameterization projects an increase on the order

of 5 DU, which could be related to the steadily increasing concentration of nitrous oxide. While the authors have nicely constructed a set of experiments to quantitatively estimate the effects of ODSs and GHGs on tropical ozone, the absence of any methodical investigation of the effects of methane or nitrous oxide would, I believe, rule out making any statements on the importance of these species.

Somewhat related to this point, it would be very helpful to the reader if the authors would state what N2O and CH4 concentrations were used for the timeslice experiments. I assume all six of the timeslice experiments used the same specifications for N2O and CH4 but it would be helpful to know if this were so and what boundary conditions specifically were used.

Aside from that my other concerns are minor and are specified below.

Page 4, Lines 22-23: Do the two extra ensemble members that start in 1980 use chemical initial conditions from the original two members that were started in 1960? If not, how are the chemical tracers for these two simulations initialized?

Page 5, Lines 2-3: In Table 1 there seems to be an error in the specifications for TS4.5_ODS as that table says climate for RCP8.5 is used.

Page 7, Lines 18-20 states 'These results indicate that over the recent past upper stratospheric ozone depletion resulting from increased Cly concentrations has in part been offset by radiative cooling resulting from increased GHG concentrations, and that in the future both increased GHG concentrations and reduced stratospheric Cly will result in increases in upper stratospheric ozone concentrations.' A very applicable reference to earlier work on this point would be Shepherd and Jonsson, On the attribution of stratospheric ozone and temperature changes to changes in ozone-depleting substances and well-mixed greenhouse gases, Atmos. Chem. Phys., 8, 1435-1444, 2008.

Page 8, Lines 8-10: 'As was seen for the upper stratosphere, the PCO3_LS response to a given decrease in ODS is dependent on the GHG concentration, (+7 DU for

TS2000_ODS - TS2000, +6 DU for TS4.5_ODS - TS4.5 and +4 DU for TS8.5_ODS – TS8.5).' Do you have any explanation for the variations in the response to ODSs across the GHG concentrations?

Page 9, Line 7. Here in reference to Figure 3 the amount of ODSs in the atmosphere is indicated by EESC. Traditionally Equivalent Effective Stratospheric Chlorine has been defined in a very particular way using tropospheric concentrations, age of air and re-lease factors for the decomposition of the ODS compound. Given the way the trace of EESC on Figure 3 looks, I think you would want to refer to Equivalent Stratospheric Chlorine (ESC). Have a look at Eyring et al., Multi-model assessment of stratospheric ozone return dates and ozone recovery in CCMVal-2 models, Atmos. Chem. Phys., 10, 9451-9472, 2010, for an example. You should also quote what value of alpha, the enhancement factor for bromine, you have used.

Page 12, Lines 12-13. The statement 'The largest rate of change for tropospheric column ozone occurs over the recent past (1960-2000) (Figure 2c), when increases in anthropogenic NOx emissions (Lamarque et al., 2010) drive increases in ozone production.' A minor point, but I do not think you can rule out the increase in methane over 1960-2000 as contributing. Methane in 1850 was ∼800 ppbv, in 1960 it was 1250 and in 2000 it was 1750 ppbv. About one-half of the total increase occurred between 1960 and 2000 and results from ACCMIP (e.g. Young et al., Pre-industrial to end 21st century projections of tropospheric ozone...., Atmos. Chem. Phys., 13, 2063-2090, 2013) show that the methane increase does account for a good portion of the total increase between 1850 and 2000.

Page 14, Lines 6 and 7: I had trouble reading 'These scenarios include RCP4.5, RCP8.5, RCP6.0 using ODS fixed at 1960 values and RCP6.0 using CDE fixed at 1960 values.' It took a bit of rereading and looking at Figure 7 to understand that not all of RCP4.5, RCP8.5 and RCP6.0 were run using ODS fixed at 1960 values. Is it possible to reword a bit.

Page 14 Lines 6 and 7: The RCP4.5 and 8.5 results from the parameterization could be compared with Figure 6 of Eyring et al., Long-term ozone changes and associated climate impacts in CMIP5 simulations, J. Geophys. Res., 118, 5029-5060, 2013. They show that going towards 2100, it is actually RCP6 that has the lowest stratospheric column ozone while RCP8.5 is slightly higher. Not to beat on this point too much, but I think the different relative order shown by your parameterization may be due to ignoring the effects of CH4. Of course, it is a different set of models compared with your parameterization derived from UM-UKCA and that cannot be ignored either.

Page 32 – Figure 6. I may have missed it, but I did not find any discussion of Figure 6 in the text.

---

## Author Comment (AC1) · 4 Aug 2017

*We thank both referees for their positive and constructive comments. Our detailed response is given below (in bold italics). Page and line numbers refer to the updated manuscript.*

*Response to Anonymous Referee #1*

General comments:

Keeble et al. examine tropical ozone trends between 1960-2100 in an ensemble of chemistry-climate model simulations following RCP 6.0. They examine trends in the upper stratosphere, lower stratosphere and troposphere, and use a set of sensitivity simulations to quantify the chemical effects of CFCs, and the radiative effects of greenhouse gases (CO2 + N2O + CH4 + CFCs). They have laid the foundation for a thorough analysis of projected tropical ozone trends, which will be of interest for the stratospheric ozone community, however I do have a number of issues with the paper in its present form that I think should be addressed before the paper is published in ACP.

The authors do not include the chemical effects of CH4 and N2O in their sensitivity simulations. As the authors themselves note (P3L14-16): "the atmospheric concentration of these species, and by extension future concentrations of HOx and NOx radicals, is therefore highly sensitive to assumptions made about their future emissions." I would have thought this a good reason to include them in the analysis, particularly as CH4 and N2O are not currently regulated, unlike the CFCs. I also do not agree with statements such as (p.13): ". . .we showed that future changes in tropical stratospheric column ozone are driven primarily by changes in: (i) the halogen-catalysed loss; (ii) the strength of tropical upwelling; and (iii) the upper stratospheric cooling induced by GHGs (mainly CO2)." You did not look at changes in N2O and CH4 chemistry, so how can you say that they are not important drivers? Or, that "the changes in HOx and NOx chemistry resulting from future changes in CH4 emissions would appear to be of second order on the timescales considered" (P13L27-28). A number of studies show that stratospheric ozone is controlled by CO2, CH4 and N2O in the second half of the 21st century (see e.g. Butler et al. 2016 and references therein), and I think the authors need to address this. N2O is the most important ODS currently emitted (Ravishankara et al., 2009), and while reductions in CFCs and increases in CO2 will have a major effect on ozone this century, I am skeptical that N2O can be considered of secondary importance, especially since its chemical effects were not included in this analysis.

*The paper has been amended to make clear that our analysis considers only the radiative effects of GHGs and the chemical effects of halogenated ODS species. Where appropriate we have added to the text discussion on the role of N2O and CH4 in chemical ozone depletion and links to tropical column ozone. Our key rationale for claiming that ODS and the radiative effects of GHGs are the key drivers of tropical stratospheric column ozone values comes from the ability of the simple model to reproduce with a reasonable degree of accuracy the long-term trends in stratospheric column ozone from 1960-2100 as modelled by the transient CCM simulation. While differences between the transient simulation and simple model may result from not including (among other things) CH4 and N2O, the very fact that the simple model is able to reproduce the main features of modelled SCO3 values from the CCM (i.e. rapid ozone loss in the late 20th century, a minimum around year 2000 and a gradual increase throughout the 21st century) highlights the important role ODSs and GHGs play in determining future ozone trends. Nevertheless, as highlighted by the reviewer and as we now discuss in the text, the simple model does not quantitatively reproduce SCO3 changes in all periods, and we now include a more balanced discussion about the limitations of the simple model.*

I would also like to see a fuller discussion of how the authors' results compare with existing chemistry-climate model studies. For example, they could be compared with the sensitivity studies of Butler et al., 2016; Eyring et al., 2010; Fleming et al., 2011; Oman et al., 2010 and Revell et al., 2012; full citations are given below. Do the authors' results confirm results from existing studies? Do they show something new?

***Fuller discussion of how our results from both the transient CCM simulations and the simple model compare with the existing literature has been added throughout the manuscript where appropriate.***

As well as comparing the results with other model studies, I think the authors should compare their RCP 6.0 simulation with observations where available, to show how well their CCM performs in the tropics.

***Merged observational data from the Bodeker ozone dataset has been used to add observed total column ozone anomalies relative to the year 2000±5 to Figure 1. A comparison between the model and the observations has been added to the text (P7L7).***

I am not convinced that the simple model discussed in Section 5 is reliable. It shows (Fig. 7) that stratospheric ozone abundances at the end of the 21st century are higher in RCP 4.5 than they are in RCP 6.0, which is higher again than in RCP 8.5. This is in direct contrast to results from existing CCMs, which show that ozone is highest in RCP 8.5 > RCP 6.0 > RCP 4.5 (see e.g. Fig. 2-23 from Chapter 2 of the WMO 2014 Ozone Assessment). And why does ozone decrease over time when ODSs are held fixed – surely GHG-induced stratospheric cooling should cause ozone to increase? C.f. e.g. Fig. 6 from Fleming et al. (2011).

***As the reviewer states, and as discussed in the reply to major comment 1, the simple model does not include a number of processes, chief amongst them the chemical effects of CH4 and N2O, which are likely to contribute to SCO3 trends in the different RCP scenarios. Differences between the multi-model results of Eyring et al (2013) and our study may also result from differences in the chemical and radiative sensitivity of the UM-UKCA model to halogenated ODS and GHGs (e.g. in the sensitivity of the BDC to changes in CDE), as discussed in the manuscript (P17L13).***

***Note that Fig. 7 is not directly comparable to Fig. 2-23 from Chapter 2 of the WMO 2014 ozone assessment which shows \*total\* column ozone. This includes tropospheric ozone changes, which overwhelm the stratospheric ozone response. We instead include a comparison of Fig. 7 to Fig. 6b in Eyring et al., 2013 in the text (P17L4) along with a discussion on the end of century SCO3 values in the simple model and compared with other studies, as the reviewer suggests (P17L8). We have caveated our discussion to highlight that the chemical effects of future CH4 and N2O changes are not included in the simple model and that this may account for some of the discrepancy between the simple model and CCM projections by the end of the 21st century and also in the relative projections from the simple model for different RCPs.***

***Stratospheric ozone decreases over time when CDE increases but ODSs are held constant (cyan line Figure 7) because the reduction in lower stratospheric ozone concentrations due to transport (due to a strengthening BDC) is able to more than compensate for increases in upper stratospheric ozone (due to $CO_2$-induced cooling) in our simulations. This point is discussed and quantified in the manuscript (section 4.2).***

The figures are generally well presented. I do have some ideas for splitting them up and recombining the various subfigures to improve the flow of the discussion (noted later on). The tables contain a few errors, which I have also noted later on in this review.

***These points are discussed below***

Specific comments

- The authors repeatedly refer to ozone recovery and "super-recovery." I understand what they are referring to, however the terminology is not correct. Ozone is projected to increase through the 21st century because (i) CFCs decrease; (ii) GHG-induced stratospheric cooling (mostly by CO2) increases. Any ozone increase induced by (ii) is not a "recovery," because it was CFCs that caused late 20th century ozone depletion in the first place. I would prefer that such statements surrounding recoveries and super-recoveries are worded more carefully.

***Where appropriate reference to recovery has been replaced with the phase 'return to 1980s values' or words to this effect.***

- Reactive chlorine is referred to as Cly and ClOx. It would improve readability if one term was used consistently.

***We thank the reviewer for pointing this out and have corrected references to Cly (which refers to inorganic chlorine species) and ClOx (=Cl + ClO i.e. active - ozone destroying - chlorine species).***

- Be careful when referring to ODS-driven ozone loss. Here you refer to ODSs (in your timeslice simulations) as Cly+Bry containing species, and do not include N2O, which is also an ODS.

***Where appropriate we have specifically referred to halogenated ODS in the introduction, and in the methodology section of the paper now highlight that we refer only to the halogenated ODS throughout the discussion of our results (P6L17).***

- The discussion of partial column ozone differences (3.2) is difficult to interpret since the drivers of ozone change are given only in the following section. The discussion of drivers of ozone change needs to come sooner. I suggest splitting up figure 2, and combining fig. 2a with figs. 3 and 4; combining fig. 2b with fig. 5; and combining fig. 2c with zonally-resolved plots (discussed later in this review). Then the partial column differences and their drivers in each region of the atmosphere can be discussed sequentially.

***We have taken this suggestion under consideration. However, we believe that it is of foremost importance to first identify the trends in total and partial column ozone (particularly with a focus on whether ozone in each region is projected to return to 1960s values), and then to later discuss the mechanisms behind these changes in the subsequent section. We feel that this subdivision into model projections and then mechanisms driving those changes best conveys the key findings of the paper. We have added a line in the introductory paragraph of Section 3 (P7, L3) which states that a detailed description of the mechanisms driving the changes presented in the section will be explored in Section 4.***

- In the introduction, you discuss the benefits of the stratospheric ozone layer for human health, however a discussion of the harmful effects on tropospheric ozone (as an air pollutant and GHG, and its negative effects on visibility and crop damage) is missing.

***A discussion of tropospheric ozone, its key drivers and its important role in air quality and as a GHG has been added to the introduction.***

- P2L1-2: At first this reads like a contradiction. The authors need to explain that in the tropics there is a small stratospheric ozone column with a high ozone concentration, and a large tropospheric ozone column with a low ozone concentration, because of the higher tropopause.

***The influence of the altitude of peak ozone mixing ratios and high tropopause height on the low tropical column values has been added to the text (P2L6).***

- P2L3-4: You might also want to mention projected changes in tropospheric ozone precursors from developing countries.

***The focus of this paper is on the long-lived radiative and chemical drivers of total column ozone, with an emphasis of the contribution from stratospheric ozone to TCO, and for this reason we do not discuss in detail the other drivers of ozone concentrations, such as tropospheric ozone precursors, which have only a small impact on stratospheric ozone concentrations (see e.g. Banerjee et al., 2016).***

- P2L10-11: note the time period these studies looked at: they show that tropical TCO3 might not reach pre-1980s values by the end of the 21st century.

***Specific reference to the fact these projections go to the end of the 21$^{st}$ century has been added to the text (P2L15).***

- P2L18: See e.g. Solomon et al., 2016.

***We thank the reviewer for drawing our attention to this paper – it has been cited in the text and added to the reference list (P2L25).***

- P3L28: While emissions to date indicate that RCP 8.5 is "business-as-usual" at present, I am uncomfortable referring to RCP 8.5 in this way since the methane concentrations by the end of the 21st century are so extreme.

***Reference to business-as-usual has been removed from the text***

- P4L16: "WCRP/SPARC" -> "IGAC/SPARC" also the reference Eyring et al. (2013) should be changed to Morgenstern et al., 2017. You could also include a sentence describing what CCMI is.

***While the Morgenstern et al. (2017) paper provides an overview of the models participating in the first phase of CCMI, Eyring et al. (2013) provide the original and detailed description of the CCMI Reference simulations and is the correct paper to cite here. Furthermore, CCMI itself is well defined within that reference. As we are not conducting a multimodel comparison we feel this manuscript is not the place to add a detailed description of the CCMI project.***

- P4L17: Was the chemistry scheme UKCA or CheS+? How are they different? Please provide more details here.

***The UKCA chemistry module is available in several possible configurations. We have used two configurations in this study: 1) the transient simulations are run in a configuration with detailed stratospheric chemistry but simplified tropospheric chemistry and 2) the time-slice simulations are run in a configuration with a coupled***

*stratospheric-tropospheric chemistry scheme. To avoid confusion, we have removed the abbreviations that describe these two UKCA configurations (CheS+ and CheST) and have instead described them in words. Please see Sect. 2 of the final manuscript for these changes.*

- P4L23-24: How were the initial atmospheric conditions perturbed for each ensemble run?

*Initial conditions for each simulation were generated from perpetual year simulations, with each ensemble member initialised from different years of this perpetual run.*

- Table 1 has some errors. I think TS4.5_ODS is supposed to read: climate = 2100 (RCP 4.5) and for TS8.5 climate = 2100 (RCP 8.5).

*We thank the reviewer for highlighting the errors in the table – these have been corrected.*

- The ODS scenarios developed for the RCPs are all rather similar and similar to the WMO A1 scenario for halocarbons, is this correct? You may want to include some detail here and thus justify why you use Year 2100 ODSs from RCP 4.5 in you TS8.5_ODS simulation.

*As the reviewer states, all RCP scenarios have very similar surface concentrations for ODS, which makes the choice of emissions scenario somewhat arbitrary for the future ODS loading time-slice experiments. We have added text to state this (P5, L30).*

*Furthermore, it is important when conducting a process-based study, as this paper does, to ensure that each perturbation is consistently applied – i.e. that the ODS emission change is identical for each pair of differences. For this reason, we consistently use the RCP4.5 scenario for 2100 ODS concentrations.*

- It would be helpful in Table 1 to note that changes in ODSs (Cly and Bry species but not NOx) are imposed only on the chemistry scheme while changes in GHGs (incl. CFCs) are imposed only on the radiation scheme.

*This clarification has been added to the table caption.*

- For experiment TS2000 do GHGs (i.e. CH4 and N2O) influence chemistry? I think so as this is your "base" run and the other five timeslice experiments are the perturbation experiments, is that correct?

*$CH_4$ and $N_2O$ affect both chemistry and radiation in all the time-slice simulations. However, only the radiative effects of changes to $CH_4$ and $N_2O$ (alongside $CO_2$ and CFCs) are considered in TS4.5 and TS8.5. In effect, all 6 time slice experiments are run with year 2000 concentrations of CH4 and N2O in the chemistry scheme, and either year 2000 or 2100 concentrations in the radiation scheme, depending on which time-slice is considered.*

- P5L18: Stating that a full description of the simulations is available in Banerjee et al. (2014) is not very helpful as they use a different nomenclature. Please include all relevant details here.

*Additional description has been added to the methodology section of the paper to enhance clarity for the reader (section 2).*

Figure 1:

- I would like to see some evaluation and discussion of how your CCM performs compared to observations; maybe by plotting observations for when they are available on Figure 1.

***Merged observational data from the Bodeker ozone dataset has been added to Figure 1 and a discussion included in the text (Sect. 3.1).***

- I am surprised that tropical total column ozone increases by so much in the mid-21$^{st}$ century (Fig. 1), and would like to see more discussion on this, as it is somewhat at odds with the existing literature (see e.g. Fig.2-23 of the WMO 2014 Ozone Assessment, Chapter 2; Fig. 6 of Eyring et al. (2013)). Is the upper stratospheric cooling in the model excessive? Or is too much ozone produced in the troposphere, for example?

***We have added to the discussion of Figure 1 a few sentences highlighting the fact that in our simulations TCO3 values do return to pre-1980s values for some part of the 21$^{st}$ century, although other studies (WMO [2014], Meul et al., 2016) suggest this is far from certain (P7L17). Note that TCO3 at 2100 is only ~2 DU lower than in 1980 (following RCP6.0) in Meul et al., 2016, which is more in line with our study. We do not compare our Fig. 1 showing TCO to Fig. 6b of Eyring et al., 2013, which only shows the stratospheric column.***

- I am missing a discussion of why TCO3 in the TS2000_ods experiment is so high (higher than in 1960 in the RCP 6.0 simulation). I think this could be because there is very little Cly-induced ozone loss, together with a strong radiative effect from GHGs, which cools the upper stratosphere and thus increases ozone – please discuss this.

***Several effects are neglected by comparing the ozone anomalies in any pair of time-slice simulations (such as TS2000 and TS2000_ODS) to anomalies in the transient runs (such as 1960-2000) e.g. the impact of changing GHGs, particularly the cooling effect of CO2, but also the role of CH4 and N2O changes, the role of the solar cycle and different aerosol loadings and SST configurations. From Figure 2c, it is clear that the smaller TCO3 value in the transient run at 1960 compared to TS2000_ODS can in large part be accounted for by an approximately 5 DU reduction in tropospheric partial column values between 1960-2000, the drivers of which are not included in the TS2000_ODS simulation. In contrast, the values at 1960 and in TS2000_ODS are much more similar for the partial stratospheric columns (particularly PCO3_LS) since most of the change in stratospheric column ozone between 1960-2100 is driven by the effect of ODSs.***

- P6L23-24: Please be more explicit here. Ozone-destroying chlorine chemistry is temperature-dependent, therefore slows in a colder stratosphere, therefore ozone increases.

***We have removed the explanation of this non-linearity here and explained in further detail in Sect. 4.***

- P7L3-4: Was there a particular reason that you chose 30 km to differentiate between the upper and lower stratosphere? Please also state the pressure level.

***30 km was chosen as the approximate altitude region where ozone changes from being predominantly under photochemical control in the upper stratosphere and predominantly under dynamical control in the lower stratosphere. The corresponding approximate pressure level has been added to the manuscript (P8L10).***

- P7L8-11: State why ozone increases, i.e. the GHG-induced stratospheric cooling effect.

*This has been added (P8L19)*

- P712-13: But as already stated by the authors, the effect of Cly forcing is non-linear and dependent on the climate scenario. So what does it mean to say that a 5 DU increase in ozone can be attributed to Cly over the 21st century, given that you are looking at a year 2000 climate? I think you're getting at that if ODS concentrations in 2000 were equal to the year 2100 values, we would expect ozone to be 5 DU higher, right?

*That is correct; we have amended the sentence to highlight the climate dependence of this value (P8L23).*

- P7L15: Please be more explicit here about what the Maycock 2016 paper shows – it looks as though you cite it to back up the statement that stratospheric cooling is GHG scenario dependent, but this has been known for a long time.

*The specific reference for the effect of CO2 on stratospheric cooling has been replaced with Manabe and Wetherald (1975) and Shine et al. (2003) (P8L26).*

- P7L22-23: Are you referring to the difference between the blue circle and triangle, and the difference between the red circle and triangle? It is hard to read from the figure, but looks like it is ~5 DU for each. That is indeed interesting – it implies that in the upper stratosphere, the climate scenario has little effect on Cly-induced ozone destruction? Why would that be?

*We have specifically highlighted the runs we are comparing in the text for clarity (P9L5).  While there is some effect of the upper stratospheric climate on the 5DU value, the non-linearity occurs above the level of the stratospheric ozone maximum (see Fig. 3 in Banerjee et al., 2016) and thus causes a small, non linear component effect in the column.*

- P8L2: "compare red/blue circles with green circle in Figure 2b" – this sort of statement is useful in interpreting the figures, and I encourage the authors to use more of them.

*Where appropriate we have used this terminology to improve the clarity of our discussion.*

- P8L6-8: this sentence is confusing; please reword it. Namely, what are you comparing to the upper stratosphere?

*This sentence has been reworded to aid clarity.  It now reads "While increases in both GHGs and stratospheric $Cl_y$ have acted to decrease $PCO3_{LS}$ in the past, in the future the effects of decreasing stratospheric $Cl_y$ and increasing GHG concentrations will have competing effects on $PCO3_{LS}$. This is in contrast to the upper stratosphere where future decreases in halogenated ODS and increases in GHG concentrations are both projected to lead to higher ozone concentrations."*

- P8L12-14: Why? Evolution of ozone precursor emissions in RCP 6.0 due to countries cleaning up their air quality?

*It is beyond the scope of this paper to explore the processes driving the tropospheric partial column trends in the transient runs in any detail given that the time-slice experiments do not consider this perturbation and the tropospheric chemistry scheme in the transient experiment being less detailed than in the time-slice experiments.  All we can say categorically based on our results is that this decrease is driven by neither ODS nor radiative changes.  While ozone precursor emissions are*

*a possibility, there exist several potential drivers.  More work would be required to explore this issue in the future.*

- P8L15: How do your results compare with the ACCMIP models? (Young et al., 2013).

*This sentence refers to the influence of GHGs (and climate) on tropical tropospheric column ozone only. A comparison to Young et al. [2013] is not possible since that study considered the combined influence of all forcings (including e.g. the chemical impacts of a large increase in $CH_4$ concentration). Moreover, as discussed above, the focus of this paper is on the stratospheric processes and drivers of total column changes given the simplified tropospheric chemistry of the transient simulations. As such, we do not compare the tropospheric trends with other models.*

- P816-17: You might want to state that this is expected because ODSs are photolysed in the stratosphere, not the troposphere.

*We have modified the sentence to clarify this point (P10L1).*

- Table 2: how are the contributions to ozone destruction calculated?

*Chemical ozone loss rates are calculated by diagnosing fluxes through each of the ozone destroying cycles included in the model and grouping them by family (e.g. $ClO_x$, $HO_x$, etc.) following the method of Lee et al. (2002) in which the rate of odd oxygen destruction is estimated for different catalytic cycles by determining the rates of their rate-limiting steps.  This has been added to the manuscript (P10L21).*

- Table 2: You show $NO_x$ and $HO_x$-induced ozone destruction, although chemical changes in $N_2O$ and $CH_4$ were not included in simulations TS2000_ODS, TS4.5 and TS8.5. . . I think you should state this in the table caption to make it clear that any changes in their rates are radiative effects or buffering by Cly.

*This point has been added to the table caption*

- P9L17: State how much of a reduction in EESC induces an increase in PCO3_US by 5 DU.

*This has been added to the text.  The sentence now reads "Comparison of $TS2000_{ODS}$ with TS2000 isolates the effects of future changes in ODSs on $PCO3_{US}$; as discussed in Section 3.2, we find that reductions in ESC from year 2000 values to projected values for year 2100 increase $PCO3_{US}$ abundances by 5 DU (8%)."*

- P9L19-20: $O_x$ loss through reactions with $O_x$? Rather the Chapman cycles?

*We use this convention to be consistent with $ClO_x$, $HO_x$ and $NO_x$.  Furthermore, the Chapman cycle refers to a set of reactions which produce, destroy and inter-convert $O_x$.  We refer only to the reaction of $O+O_3$, the reaction of $O_x$ with $O_x$ which leads to destruction of 2 $O_x$ molecules.*

- P9L22: The upper stratosphere warms when GHGs are held constant but Cly is decreased from 2000 to 2100 concentrations. Please clarify this.

*This point has been clarified in the manuscript (P11L8).  The sentence now reads "The upper stratosphere warms by ~2 K (Figure 4) when GHGs are held constant but ODS concentrations are reduced from year 2000 to year 2100 concentrations, consistent*

*with the effect of increasing ozone concentrations on upper stratospheric temperatures as discussed by Maycock (2016)."*

- P9L24: But as well as temperature effects, HOx and NOx cycles will also be buffered by interactions with Cly. This should also be discussed.

*This point has been added to the text, which now reads "Reactions involving HOx and NOx have weaker temperature dependencies and are coupled to Cly concentrations through null cycles and the formation of reservoir species, and thus they show smaller increases."*

- P10L1-18: As mentioned earlier, it would be great if the discussion of ozone drivers came earlier.

*As discussed above, we feel the clearest presentation of the results in the paper are to sow the total column projections for the UM-UKCA model, subdivide these changes into the partial column changes to identify key differences between the different altitude regions, and then discuss the mechanisms driving these changes*

- Figure 5c is not discussed in the text.

*This has been rectified in the text (P12L17, P13L2).*

- P11L12-14: How are non-linearities accounted for here?

*It was found that a linear fit through the data points used for this calculation had an $r^2$ value of 0.96, and there was no evidence of any consistent non-linearities, and so they are not considered here.*

- P11L16-17: CO is also an important ozone precursor.

*CO has been added to the list of ozone precursors (P13L24).*

- Figure 2c: In the tropical troposphere, different chemistry regimes are at play, and a lot of information can be lost through zonal averaging. For example, in the tropical Western Pacific region ozone loss via the H2O + O(1D) reaction is very important where solar actinic fluxes and humidity are high. However in other regions, ozone production can dominate due to anthropogenic emissions of ozone precursors (biomass burning etc). I think it would be interesting to somehow resolve figure 2c zonally, and discuss a bit more the chemical changes happening there.

*The reviewer makes an important point about the regional effects of short-lived species that affect tropospheric ozone. However, the focus of this study is on the radiative and chlorine drivers of ozone concentrations resulting from changes in long-lived GHGs and halogenated source gases. Assessing zonal asymmetries in the drivers of the tropospheric ozone burden is therefore beyond the scope of this study.*

- P12L8-9: State where this is shown (Fig. 2c).

*This has been added to the text. The sentence now reads "While reductions in ODS affect tropospheric ozone in the extratropics through STE (e.g. Banerjee et al., 2016), in the tropics, ODS have little impact on tropospheric ozone, with $PCO3_T$ increasing by <1 DU in the $TS2000_{ODS}$ experiment compared to TS2000 (see Figure 2c)."*

- P12L11: I would argue that ozone precursors are a major consideration, rather than an additional consideration. . . I think you could look at their effects here too, as from Banerjee et al. (2016) I understand you have simulations available where climate and ozone precursor emissions are perturbed separately and together?

*As discussed above, given the simplified chemistry scheme of the transient UM-UKCA simulations and the focus of this paper on the long lived GHGs and halogenated ODS, we feel that further consideration of ozone precursors is beyond the scope of this paper*

- How were ozone precursor emissions prescribed in your timeslice simulations? The same as RCP 6.0?

*Ozone precursor emissions are prescribed to their year 2000 values in all of the time-slice simulations. Thus, we have not isolated the impact of a perturbation to ozone precursor emissions to year 2100 (RCP6.0) values, since the focus of this study is on the impact of climate and ODSs on column ozone. However, in Sect. 4.3 (final paragraph), we use previous studies [e.g. Revell et al., 2015] which have isolated the impacts of ozone precursors to qualitatively infer their likely role in the evolution of the tropospheric ozone column in our transient simulations.*

- P12L17: Also compare with the ACCMIP models in Young et al. (2013).

*This reference has been added to the manuscript (P14L22).*

- P14L6-7: was CDE fixed or CO2? In the text you say that CDE was fixed, but in the legend on Fig. 7 it says that CO2 was fixed. Please use consistent terminology. I think too that the caption for Fig. 7 should provide a description of the experiments shown.

*CDE was fixed as multiple GHGs are considered.  The legend in Figure 7 has been corrected to reflect this.*

- Figure 7: Why does the simple model overestimate ozone loss between ~1990-2070?

*As discussed in the replies to major comments 1 and 4, the simple model, by definition, does not capture all the processes that affect ozone in the transient simulation. While the simple model does overestimate the minimum SCO3 values occurring around year 2000, partly owing to the effects of coincident solar maximum conditions that affect the transient runs but are not included in the simple model, the rate of increase from 2000 to 2040 is similar between the simple model and the transient simulation, highlighting the importance of ODS and CDE changes to projected stratospheric ozone increases in the first half of the 21$^{st}$ century.  As we state in the paper, the simple model is not intended to replace fully coupled chemistry climate simulations, but rather to act as a framework for identifying the key drivers of future SCO3 changes.  The simple model could be further expanded to include additional terms to represent additional processes which would be likely to improve its quantitative fidelity, and it is hoped that this can be done in the future.  However, here we present only an assessment of a simple model constructed using terms we had available given the simulations used in this study and compare it to the fully coupled model to highlight i) that the broad trend of the transient simulation is reproduced in a simple, 2 component model, and ii) there remain differences between the two models highlighting the importance of other compounds and processes.*

- Discussion of fig. 7: Non-linearities are not discussed; (Meul et al., 2015) may provide helpful background information here.

*The text has been amended to include a discussion of non-linearities*

- P15L7: you are talking in terms of the total column, right? Again, I am missing a discussion of the role of tropospheric ozone as an air pollutant – even if lower stratospheric ozone losses are balanced in the total column by tropospheric increases, the result is not great for life in the biosphere because of reduced stratospheric ozone shielding the biosphere from UV-B radiation, and increased tropospheric ozone acting as an air pollutant and GHG.

*This has been added to the text (P18L8)*

- P15L8-10: Again, I disagree since these were the only factors you looked at, so you cannot discount other factors.

*The conclusions of the paper has been amended to highlight that we do not consider here the chemical effects of CH4 and N2O and discuss the role they may have on total column ozone.*

- P15L28-30: This was not discussed earlier, please include this discussion in the results section.

*This is discussed in the final paragraph of section 4.2 of the manuscript*

- Please state where your data are available from.

*The transient simulations are available as part of the CCMI initiative through BADC. Any further data are available upon request. We have added a short section on page 29 to state the availability of data.*

Technical corrections

- P1L18 "significant differences to" -> "significant differences in"
*Corrected*

- P2L6: Montreal Protocol and its subsequent amendments -> Montreal Protocol and its subsequent Adjustments and Amendments
*Corrected*

- P2L21: "over the course of the 21st century perturb" -> "over the course of the 21st century are expected to perturb"
*Corrected*

- P2L23-24: CFCs are source gases for Cly, N2O is a source gas for NOx and CH4 is a source gas for HOx. Please phrase this more carefully.
*Corrected*

- P2L26: "increases to the rate constant" -> "increases in the rate constant"
*Corrected*

- P2L27: "decreases to the rate constant" – as above.
*Corrected*

- P3L6: define Cly and NOy.
*Cly and NOy have been defined when first used*

- P3L24 onwards: there is no need to refer to "RCP emissions scenarios" or "RCP scenarios." Calling them RCPs is sufficient.
*For clarity we prefer to refer to them as RCP scenarios and so do not feel the text needs to be changed.*

- P3L28: "rise" -> "increase"
*Corrected*

- P5L3: "integration given" -> "integration are given"
*Corrected*

- P6L16: "discussed in" -> "discussed by"
*Corrected*

- P7L14: "century is dependent" -> "century are dependent"
*Corrected*

- P8L5-6: units are in italics.
*Corrected*

- P9L3: "62 DU" – it says 63 DU in Table 2.
*Corrected*

- P11L18: NOx: fix subscript.
*Corrected*

- P11L30: "increase in LNOx at RCP 8.5" -> "increase in LNOx in RCP 8.5"
*Corrected*

- P12L15: Meinhausen -> Meinshausen
*Corrected*

- P13L26 "emissions of GHGs" -> "the radiative effects of GHG emissions"
*Corrected*

- P13L28: dynamic -> dynamical
*Corrected*

- P15L3: troposphere height -> tropopause height
*Corrected*

---

## Author Comment (AC2) · 4 Aug 2017

*We thank both referees for their positive and constructive comments. Our detailed response is given below (in bold italics). Page and line numbers refer to the updated manuscript.*

*Response to Anonymous Referee #2*

The manuscript presents an investigation of the model-projected evolution of ozone in the tropics (10S to 10N) over the period 1960 to 2100. A transient simulation covering the whole period and following the specified reference scenario for the Chemistry-Climate Model Initiative (CCMI) model intercomparison project, the REF-C2 simulation, is augmented with time-slice simulations for year 2000 and 2100 conditions run under different levels of ozone depleting substances. The influence of changing greenhouse gases (GHGs) and ozone depleting substances (ODSs) on tropical total column ozone are investigated by splitting the total column into upper stratospheric, lower stratospheric and tropospheric components. Linear functions of the change in column ozone due to the effects of GHGs and ODSs are derived and these functions are used to reproduce the evolution of stratospheric ozone column in the full model simulation.

Of significance, the results add to a number of recent papers that underline the importance of ODS-driven changes in ozone on tropical upwelling in the lower stratosphere. The core of the methodology and results presented in the manuscript are, in my opinion, solid. My one significant concern is the way in which the effects of methane and nitrous oxide are treated.

On page 13, Lines 1-4, the authors state:

'In Section 4 we showed that future changes in tropical stratospheric column ozone are driven primarily by changes in: (i) the halogen-catalysed loss; (ii) the strength of tropical upwelling; and (iii) the upper stratospheric cooling induced by GHGs (mainly CO2).' I would argue that the authors have not, in fact, shown this in general. The inferred causes of changes in partial column ozone are derived from the set of timeslice experiments that only varied GHGs and ODSs. That these are then the only two factors that were found to be responsible for changes in ozone should naturally follow.

On Page 5, Lines 9-11 the authors state 'In this study we consider the radiative impact of a large number of GHG species (CO2, CH4, N2O, CFCs) and assume that the dominant driver of chemical changes is changes to ODS loadings. In this way, the chemical impact of changing N2O and CH4 emissions is not considered here.' Since the effects of changing N2O and CH4 are not considered it seems difficult to justify the conclusion (Page 16, Lines 21-23) that 'Results from the simple model indicate stratospheric column ozone changes resulting from future CH4 and N2O emissions are of second order on the timescales considered here.'

While the parameterization of stratospheric column that is derived here is able to reproduce fairly well the evolution of stratospheric column in the transient simulation, the variation of methane is fairly small in RCP6. The parameterized stratospheric column also significantly overestimates the trend from 2020 to 2100, where the full model shows almost no change while the parameterization projects an increase on the order of 5 DU, which could be related to the steadily increasing concentration of nitrous oxide. While the authors have nicely constructed a set of experiments to quantitatively estimate the effects of ODSs and GHGs on tropical ozone, the absence of any methodical investigation of the effects of methane or nitrous oxide would, I believe, rule out making any statements on the importance of these species.

*We agree that the role of future CH4 and N2O emissions for tropical TCO trends should not be understated. We have added text throughout the manuscript to this effect, and have amended the discussion in section 5 so that it does not appear that we are saying CH4 and N2O are not important drivers of future ozone projections.*

*The reason for claiming, as we do in Section 4, that 'future changes in tropical stratospheric column ozone are driven primarily by changes in: (i) the halogen-catalysed loss; (ii) the strength of tropical upwelling; and (iii) the upper stratospheric cooling induced by GHGs (mainly CO2)' is because the simple model, which includes only terms for chemical ODS and radiative CDE forcings, is able to reproduce with a reasonable degree of accuracy the long-term SCO3 trend from the fully coupled CCM simulation, which also includes CH4 and N2O chemical effects in its detailed chemical scheme. However, we do not know, for example, whether in the UKCA model, N2O and CH4 have very large but opposite effects on stratospheric ozone that cancel each other out. There is some indication from the latest WMO assessment that CH4 and N2O changes have opposite effects on column ozone, and further work with a number of fully coupled chemistry climate models is required to fully understand the drivers of future ozone concentrations in different regions of the atmosphere. We have expanded where appropriate our discussion on the roles of N2O and CH4 and the impacts that not including them in the simple model may have. We have further discussed the role of N2O and CH4 for determining differences in future SCO3 trends between RCP scenarios and the limitations therefore of the simple model in capturing details of the differences between RCPs (e.g. compared to the multi-model results of Eyring et al. (2013)).*

Somewhat related to this point, it would be very helpful to the reader if the authors would state what N2O and CH4 concentrations were used for the timeslice experiments. I assume all six of the timeslice experiments used the same specifications for N2O and CH4 but it would be helpful to know if this were so and what boundary conditions specifically were used.

*In all time-slice experiments chemical concentrations of $N_2O$ and $CH_4$ use prescribed year 2000 concentrations from RCP6.0 as a lower boundary condition. This information has been added to the manuscript (P6L3)*

Aside from that my other concerns are minor and are specified below.

Page 4, Lines 22-23: Do the two extra ensemble members that start in 1980 use chemical initial conditions from the original two members that were started in 1960? If not, how are the chemical tracers for these two simulations initialized?

*The two extra ensemble members started from 1980 use initial conditions generated by spinning off a 1980 timeslice from one of the ensemble members, which was run for 20 years, New transient simulations were then initialised using different years from this time-slice. This description has been added to the methodology section of the manuscript.*

Page 5, Lines 2-3: In Table 1 there seems to be an error in the specifications for TS4.5_ODS as that table says climate for RCP8.5 is used.

*The RCP scenarios used for the climate component of each of the experiments has been corrected.*

Page 7, Lines 18-20 states 'These results indicate that over the recent past upper stratospheric ozone depletion resulting from increased Cly concentrations has in part been

offset by radiative cooling resulting from increased GHG concentrations, and that in the future both increased GHG concentrations and reduced stratospheric Cly will result in increases in upper stratospheric ozone concentrations.' A very applicable reference to earlier work on this point would be Shepherd and Jonsson, On the attribution of stratospheric ozone and temperature changes to changes in ozone-depleting substances and well-mixed greenhouse gases, Atmos. Chem. Phys., 8, 1435-1444, 2008.

***We thank the reviewer for bringing this paper to our attention - the reference has been added to the text and reference list***

Page 8, Lines 8-10: 'As was seen for the upper stratosphere, the PCO3_LS response to a given decrease in ODS is dependent on the GHG concentration, (+7 DU for TS2000_ODS - TS2000, +6 DU for TS4.5_ODS - TS4.5 and +4 DU for TS8.5_ODS – TS8.5).' Do you have any explanation for the variations in the response to ODSs across the GHG concentrations?

***The variation in the response of ozone to ODS under different GHG loadings is related to the impact of ODS on the speed of the BDC, the temperature dependence of the chlorine catalysed loss cycles and the influence of the upper stratospheric shielding on the lower stratosphere. In all simulations, decreasing ODS concentrations lead to a deceleration of the BDC, decreasing the transport of ozone out of the tropical lower stratosphere and leading to increased lower stratospheric partial column values. The deceleration of the BDC due to ODSs is relatively insensitive to the GHG loading. However, as the stratosphere cools the efficiency of the $ClO_x$ catalysed ozone loss is reduced. Although this plays a minor role in the lower stratosphere, where only a small proportion of the CFCs have been oxidised, it does contribute to the signal seen here. In addition, cooling of the upper stratosphere leads to greatly increased ozone concentrations through both Chapman chemistry and reducing the efficiency of the $ClO_x$ catalysed ozone loss. Increased overhead ozone in turn affects the photolysis rates in the lower stratosphere, slowing ozone production. Decreased production values in the lower stratosphere partially offset the increases from a slower BDC, explaining the variations in the response to ODSs across the GHG concentrations. This discussion has been added to the manuscript.***

Page 9, Line 7. Here in reference to Figure 3 the amount of ODSs in the atmosphere is indicated by EESC. Traditionally Equivalent Effective Stratospheric Chlorine has been defined in a very particular way using tropospheric concentrations, age of air and release factors for the decomposition of the ODS compound. Given the way the trace of EESC on Figure 3 looks, I think you would want to refer to Equivalent Stratospheric Chlorine (ESC). Have a look at Eyring et al., Multi-model assessment of stratospheric ozone return dates and ozone recovery in CCMVal-2 models, Atmos. Chem. Phys., 10, 9451-9472, 2010, for an example. You should also quote what value of alpha, the enhancement factor for bromine, you have used.

***Following the definitions of Eyring et al. (2007), we have used ESC rather than EESC. This has been corrected. For alpha we have used a value of 60. This has been corrected in the text.***

Page 12, Lines 12-13. The statement 'The largest rate of change for tropospheric column ozone occurs over the recent past (1960-2000) (Figure 2c), when increases in anthropogenic NOx emissions (Lamarque et al., 2010) drive increases in ozone production.' A minor point, but I do not think you can rule out the increase in methane over 1960-2000 as contributing. Methane in 1850 was ~800 ppbv, in 1960 it was 1250 and in 2000 it was 1750 ppbv. About one-half of the total increase occurred between 1960 and 2000 and results from

ACCMIP (e.g. Young et al., Pre-industrial to end 21[st] century projections of tropospheric ozone...., Atmos. Chem. Phys., 13, 2063-2090, 2013) show that the methane increase does account for a good portion of the total increase between 1850 and 2000.

***The text of the manuscript has been amended to include the role of CH4 in historic tropospheric ozone changes alongside the Young et al. reference.***

Page 14, Lines 6 and 7: I had trouble reading 'These scenarios include RCP4.5, RCP8.5, RCP6.0 using ODS fixed at 1960 values and RCP6.0 using CDE fixed at 1960 values.' It took a bit of rereading and looking at Figure 7 to understand that not all of RCP4.5, RCP8.5 and RCP6.0 were run using ODS fixed at 1960 values. Is it possible to reword a bit.

***The description of the scenarios performed using the simple model has been revised to avoid confusion.***

Page 14 Lines 6 and 7: The RCP4.5 and 8.5 results from the parameterization could be compared with Figure 6 of Eyring et al., Long-term ozone changes and associated climate impacts in CMIP5 simulations, J. Geophys. Res., 118, 5029-5060, 2013. They show that going towards 2100, it is actually RCP6 that has the lowest stratospheric column ozone while RCP8.5 is slightly higher. Not to beat on this point too much, but I think the different relative order shown by your parameterization may be due to ignoring the effects of CH4. Of course, it is a different set of models compared with your parameterization derived from UM-UKCA and that cannot be ignored either.

***We have included a comparison of the simple model results with the Eyring et al. multimodel projections to highlight the differences between the scenarios at the end of the century and also include a discussion about how these differences may be due to CH$_4$ and N$_2$O.***

Page 32 – Figure 6. I may have missed it, but I did not find any discussion of Figure 6 in the text.

***Figure 6 should have been referenced in relation to the calculation of the $\frac{\Delta SCO3}{\Delta CDE}$ and $\frac{\Delta SCO3}{\Delta ESC}$ – this has been added to the manuscript.***